# Equivariant Neural Tangent Kernels

Philipp Misof [1]  Pan Kessel [2]  Jan E. Gerken [1]

## Abstract

Little is known about the training dynamics of equivariant neural networks, in particular how it compares to data augmented training of their non-equivariant counterparts. Recently, neural tangent kernels (NTKs) have emerged as a powerful tool to analytically study the training dynamics of wide neural networks. In this work, we take an important step towards a theoretical understanding of training dynamics of equivariant models by deriving neural tangent kernels for a broad class of equivariant architectures based on group convolutions. As a demonstration of the capabilities of our framework, we show an interesting relationship between data augmentation and group convolutional networks. Specifically, we prove that they share the same expected prediction over initializations at all training times and even off the data manifold. In this sense, they have the same training dynamics. We demonstrate in numerical experiments that this still holds approximately for finite-width ensembles. By implementing equivariant NTKs for roto-translations in the plane ($G = C_n \ltimes \mathbb{R}^2$) and 3d rotations ($G = \mathrm{SO}(3)$), we show that equivariant NTKs outperform their non-equivariant counterparts as kernel predictors for histological image classification and quantum mechanical property prediction.

## 1. Introduction

Equivariant neural networks (Weiler et al., 2023; Gerken et al., 2023) are widely used in many applications of great practical importance, for example in medical image analysis in two and three dimensions (Bekkers et al., 2018; Winkels & Cohen, 2019; Müller et al., 2021; Pang et al., 2023) and in quantum chemistry (Duval et al., 2023; Batzner et al., 2022;

Schütt et al., 2021; Unke et al., 2021). Other application areas include particle physics (Bogatskiy et al., 2020), cosmology (Perraudin et al., 2019) and even fairness in large language models (Basu et al., 2023).

Recently, there has been a number of works which avoid equivariant architectures but rely on data augmentation to approximately learn equivariance, most notably AlphaFold3 (Abramson et al., 2024). This has the potential advantage that non-equivariant architectures may offer better training dynamics, for example favorable scaling capabilities. There has been a vigorous debate on this subject with some empirical works claiming superiority of equivariant architectures (Gerken et al., 2022; Brehmer et al., 2024) while others suggest the opposite (Wang et al., 2024; Abramson et al., 2024). One challenging aspect to conclusively settle the matter is that there is no good theoretical understanding of how the equivariant and the purely augmentation-based training dynamics compare.

Motivated by this observation, this paper derives equivariant *neural tangent kernel (NTK)* theory (Jacot et al., 2018) for group convolutional architectures. The NTK provides a powerful tool to analytically study the training dynamics of neural networks in the large width limit by analyzing the behavior of the kernel, in particular its trace, eigenvalues and other properties (Geiger et al., 2020; Mok et al., 2022; Engel et al., 2024; Tsai et al., 2023). A particularly important feature of the NTK is the fact that in the infinite width limit, it becomes constant throughout training (Jacot et al., 2018). Furthermore, at infinite width, the NTK can be computed by layer-wise recursion relations. These simplifications allow for complete analytic control over the training dynamics. In particular, the network output of an arbitrary input and at an arbitrary point in training time converges to a Gaussian process over initializations whose mean- and covariance functions can be computed analytically. This result has led to a number of theoretical and practical insights (Geiger et al., 2020; Jacot et al., 2020; Yang & Hu, 2021; Yang et al., 2021; Franceschi et al., 2022; Day et al., 2023) into the initialization and training of neural networks.

We derive recursive relations which determine the NTK of an equivariant neural networks for the first time. In particular, we study the NTK of group convolutional layers (Cohen & Welling, 2016). These layers are in some sense universal.

[1]Department of Mathematical Sciences, Chalmers University of Technology and the University of Gothenburg, SE-412 96 Gothenburg, Sweden [2]Prescient Design, Genentech Roche, Basel, Switzerland. Correspondence to: Jan Gerken <gerken@chalmers.se>.

*Proceedings of the 42$^{nd}$ International Conference on Machine Learning*, Vancouver, Canada. PMLR 267, 2025. Copyright 2025 by the author(s).

Specifically, they have the unique property that they arise from imposing an equivariance constraint on dense fully-connected layers and are therefore the most general linear, equivariant transformations (Kondor & Trivedi, 2018) and have been used in a wide array of applications (Chidester et al., 2019; Celledoni et al., 2021; Moyer et al., 2021).

These NTK recursion allows us to clarify the relation between the training dynamics of pure data augmentation and equivariant architectures in the large width limit. Specifically, non-equivariant architectures trained with full data augmentation converge to certain group convolutional architectures in the infinite width limit. This result holds for any input, in particular off-manifold, and at any training time. Thus, at least in the infinite width limit and in expectation over initializations, the training dynamics of data augmentation is identical to the one of certain group convolutional architectures.

NTKs have also been shown to be interesting kernel functions in their own right. Since they are induced by neural network architectures, they allow to transfer the intuition gained in the extensive literature on the design of neural networks to kernel machines and have shown to outperform more traditional kernel functions (Arora et al., 2019; Li et al., 2019; Lee et al., 2020). In our experiments, we show that group equivariant kernels outperform their non-equivariant counterparts for both regression and classification as well as for discrete roto-translations and continuous rotations. In summary, our main contributions are

- We derive layer-wise recursive relations for the neural tangent kernel and neural network Gaussian process kernel of group convolutional layers, the corresponding lifting layers, point-wise nonlinearities and group-pooling layers.

- We specialize our general results to the case of roto-translations in the plane as well as the three-dimensional rotation group $SO(3)$. We derive and implement the kernel relations for these cases, allowing for efficient computations. The code is provided publicly at `https://github.com/PhilippMi sofCH/equivariant-ntk`.

- We prove that in the infinite width limit, a standard convolutional or fully connected network trained with full data augmentation yields the same expected network function as a corresponding group convolutional network trained without data augmentation . This result holds for all training times as well as off manifold. We show empirically that this holds approximately at finite width.

- We verify experimentally that the NTKs of finite-width equivariant networks converge to our equivariant NTKs

as width grows to infinity. Furthermore, we demonstrate the superior performance of equivariant NTKs over other kernels for medical image classification and quantum mechanical property prediction.

## 2. Related Work

**Neural Tangent Kernel.** Gaussian processes can be viewed as Bayesian neural networks as first pointed out by (Neal, 1996) and this relation extends to deep neural networks as shown in (Lee et al., 2018). Neural tangent kernels allow description of training dynamics, see the seminal reference (Jacot et al., 2018) and (Golikov et al., 2022) for an accessible review. In (Lee et al., 2019), NTK theory was used to show that wide neural networks trained with gradient descent become Gaussian processes and generalized in a more rigorous and systematic manner by (Yang, 2020). NTKs can be used to derive parametrizations that allow scaling networks to large width (Yang & Hu, 2021). They can also be used to theoretically analyze GANs (Franceschi et al., 2022), PINNs (Wang et al., 2022b), backdoor attacks (Hayase & Oh, 2023), pruning (Yang & Wang, 2023) and spectral learning biases (Bordelon et al., 2020; Canatar et al., 2021). Recently, corrections to infinite width limit have been studied by (Huang & Yau, 2020; Yaida, 2020; Halverson et al., 2021; Erbin et al., 2022) using techniques inspired by perturbative quantum field theory. The NTK kernel for convolutional architectures was derived in (Arora et al., 2019). Our results can be thought as a generalization thereof to general group convolutions.

**Equivariant Neural Networks.** Equivariance has been an important theme of deep learning research over the last years, see (Gerken et al., 2023) for an accessible review. Equivariant deep learning is part of the larger area of geometric deep learning (Bronstein et al., 2017), in which more general geometric properties of the different parts of the learning problem (e.g. the data (Dombrowski et al., 2024), model (Weiler et al., 2023) and optimization procedure (Amari, 1998)) are studied. Herein, we focus on group convolutional layers (Cohen & Welling, 2016) which are the unique linear equivariant layers. A comprehensive summary is given in (Cohen, 2021). These architectures have found wide-spread application in computer vision (Chidester et al., 2019; Celledoni et al., 2021; Moyer et al., 2021), medical applications (Bekkers et al., 2018; Chidester et al., 2019; Pang et al., 2023) as well as natural science use cases (Nicoli et al., 2020; Liao & Smidt, 2023; Bekkers et al., 2024).

**Learned vs. Manifest Equivariance.** While equivariance can be enforced on the training data via data augmentation, the imposed symmetry does in general not extend to out-of-distribution data (Moskalev et al., 2023). This is in accordance with the tighter upper bound on the generaliza-

tion error of equivariant networks compared to purely data-augmented ones that was found in (Wang et al., 2022a). By analyzing the spectrum of the NTK in a toy problem, (Perin & Deny, 2024) found that non-equivariant neural networks are unable to generalize symmetries learned in one class via data augmentation to another, only partially augmented class. They further show how the qualitatively different NTK spectrum of a CNN improves generalization for the task under consideration. In (Gerken & Kessel, 2024), the effect of data augmentation on infinitely wide neural network ensembles was studied. The authors found that the resulting Gaussian process is equivariant at all training times and even off the data manifold. In contrast, our results do not require data augmentation but derive an NTK for manifestly equivariant group convolution layers. This allows us to find a connection between the ensemble means of data-augmented and equivariant networks, which complements the previously mentioned results focusing on individual networks.

## 3. Background

This section gives a brief overview of NTK theory (for an optional broader introduction, see Appendix A) as well as of equivariant neural networks with a particular emphasis on group convolutional neural networks (GCNNs).

**Neural Tangent Kernels.** The NTK can be computed by layer-wise recursive relations (Jacot et al., 2018) starting from the definition

$$\Theta^{(\ell)}(x, x') = \mathbb{E}\left[\sum_{\ell'=1}^{\ell} \frac{\partial \mathcal{N}^{(\ell)}(x)}{\partial \theta^{(\ell')}} \left(\frac{\partial \mathcal{N}^{(\ell)}(x')}{\partial \theta^{(\ell')}}\right)^{\top}\right] \quad (1)$$

of the layer-$\ell$ NTK. The NTK of the full network is given by $\Theta(x, x') = \Theta^{(L)}(x, x')$ for a network depth $L$. Here, $\theta^{(\ell)}$ are the parameters of the layer $\ell$ and we adopt the convention that expectation values are over the initialization distribution unless otherwise stated. In the limit of infinitely wide hidden layers, not only do analytic expressions exist for (1), but $\Theta^{(\ell)}(x, x')$ also stays constant during training. Hence it is often referred to as the *frozen* NTK (Geiger et al., 2020). As customary in the NTK literature, we treat activations and preactivations as distinct layers and refer to $\mathcal{N}^{(\ell)}$ as the layer-$\ell$ features with $\mathcal{N}(x) = \mathcal{N}^{(L)}(x)$. This allows us to treat linear- and nonlinear layers on an equal footing. Since (1) is proportional to the unit matrix, we can treat it as a scalar.

We can find a recursion relation between $\Theta^{(\ell+1)}$ and $\Theta^{(\ell)}$ by separating the $\ell' = \ell + 1$ contribution from the sum and computing the $\ell' \leq \ell$ contributions in terms of derivatives

through the layer $\ell + 1$ using the chain rule,

$$\Theta^{(\ell+1)}(x, x') = \mathbb{E}\left[\frac{\partial \mathcal{N}^{(\ell+1)}(x)}{\partial \theta^{(\ell+1)}} \left(\frac{\partial \mathcal{N}^{(\ell+1)}(x')}{\partial \theta^{(\ell+1)}}\right)^{\top}\right]$$
$$+ \mathbb{E}\left[\frac{\partial \mathcal{N}^{(\ell+1)}(x)}{\partial \mathcal{N}^{(\ell)}(x)} \underbrace{\left(\sum_{\ell'=1}^{\ell} \frac{\partial \mathcal{N}^{(\ell)}(x)}{\partial \theta^{(\ell')}} \left(\frac{\partial \mathcal{N}^{(\ell)}(x')}{\partial \theta^{(\ell')}}\right)^{\top}\right)}_{\Theta^{(\ell)}(x, x')}\right.$$
$$\left. \times \left(\frac{\partial \mathcal{N}^{(\ell+1)}(x)}{\partial \mathcal{N}^{(\ell)}(x)}\right)^{\top}\right]. \quad (2)$$

Note that according to the NTK's definition (1), it holds that $\Theta^{(0)} = 0$. The recursions (2) have been computed explicitly for a number of layers, e.g. fully connected (Jacot et al., 2018), nonlinear (Jacot et al., 2018), convolution (Arora et al., 2019), and graph convolution (Du et al., 2019). An efficient implementation for many layers is available in the Jax-based Python package `neural-tangents` (Novak et al., 2020).

For evaluating the expectation values in (2), it is convenient to introduce the *neural network Gaussian process (NNGP)* kernel

$$K^{(\ell)}(x, x') = \mathbb{E}\left[\mathcal{N}^{(\ell)}(x) \left(\mathcal{N}^{(\ell)}(x')\right)^{\top}\right], \quad (3)$$

whose name originates in the fact that at initialization, the neural network converges in the infinite width limit to a zero-mean Gaussian process with covariance function $K^{(L)}(x, x')$ (Neal, 1996; Lee et al., 2018). In the infinite width limit, $K$ is proportional to the unit matrix, so we will treat it as a scalar as well. The NNGP can also be computed recursively layer-by-layer. For the $\ell = 0$, the NNGP is the covariance matrix of the input features $K^{(0)}(x, x') = x\, x'^{\top}$.

Using the definition (3) of the NNGP, we can determine the structure of the NTK recursive relations from (2). For linear layers, the first expectation value will evaluate to the NNGP, while the second expectation value will be proportional to the unit matrix due to the initialization with independent normally distributed parameters. For nonlinear layers, the first expectation value vanishes and the second expectation values will depend on the derivative of the nonlinearity.

**Group Convolutions.** Group convolutions (Cohen & Welling, 2016) act on feature maps $f : G \to \mathbb{R}^{n_{\text{in}}}$ where $n_{\text{in}}$ denotes the number of input features to the network. In the example of image inputs, this feature map would be $f : \mathbb{Z}^2 \to \mathbb{R}^3$ where $\mathbb{Z}^2$ is the pixel grid, $\mathbb{R}^3$ is the space of RGB colors, and the feature map $f$ is supported on $[0, h] \times [0, w]$ for imagesize $h \times w$. Let $L^2(X, Y)$ denote the set of square integrable functions from $X$ to $Y$. The $\ell$-th neural network layer $\mathcal{N}^{(\ell)} : L^2(G, \mathbb{R}^{n_{\text{in}}}) \to L^2(G, \mathbb{R}^{n_\ell})$

maps an input feature map $f : G \to \mathbb{R}^{n_{\text{in}}}$ to an output feature map $\mathcal{N}^{(\ell)}(f) : G \to \mathbb{R}^{n_\ell}$. A particular instance of such a layer is the group convolution layer which in NTK representation is given by

$$[\mathcal{N}^{(\ell+1)}(f)](g) = \frac{1}{\sqrt{n_\ell |S_\kappa|}} \int_G \mathrm{d}h \; \kappa(g^{-1}h) [\mathcal{N}^{(\ell)}(f)](h) \,, \tag{4}$$

with filter $\kappa : G \to \mathbb{R}^{n_\ell, n_{\ell+1}}$ with support $S_\kappa \subset G$. Here, we integrate over the group with respect to the Haar measure. For finite groups, the integral becomes a sum over group elements. Due to the invariance of the Haar measure, the layers (4) are equivariant with respect to the regular representation

$$(\rho_{\text{reg}}(g)f)(h) = f(g^{-1}h) \qquad g, h \in G \,. \tag{5}$$

Since the input features typically have domain $X \subseteq \mathbb{R}^{n_{\text{in}}}$ which is not the symmetry group $G$, the first layer of a *group convolutional neural network (GCNN)* is a *lifting layer* which maps a feature map with domain $X$ equivariantly into a feature map with domain $G$ (Cohen & Welling, 2016)

$$[\mathcal{N}^{(1)}(f)](g) = \frac{1}{\sqrt{n_{\text{in}} |S_\kappa|}} \int_X \mathrm{d}x \; \kappa(\rho(g^{-1})x) f(x) \,, \tag{6}$$

where $\rho$ is a representation of $G$ on $X$. We assume here and in the following that $X$ is a homogeneous space of $G$, i.e. that any two elements of $X$ are connected by a group transformation.

As is common for other network types as well, the nonlinearities in group convolutional networks are applied component-wise across the different group elements,

$$[\mathcal{N}^{(\ell+1)}(f)](g) = \sigma\big([\mathcal{N}^{(\ell)}(f)](g)\big) \tag{7}$$

for nonlinearity $\sigma$. Due to this component-wise structure, the layers (7) are equivariant with respect to the regular representation (5) as well.

By combining lifting- and group-convolution layers with nonlinearities, one can construct expressive architectures which are equivariant with respect to the regular representation, i.e. which satisfy

$$\mathcal{N}(\rho_{\text{reg}}(g)f) = \tilde{\rho}_{\text{reg}}(g)\mathcal{N}(f) \,, \qquad g \in G \,. \tag{8}$$

Many practical applications necessitate an invariant network

$$\mathcal{N}(\rho_{\text{reg}}(g)f) = \mathcal{N}(f) \,, \qquad g \in G \,. \tag{9}$$

Such a transformation property can be achieved by appending a *group pooling layer* to a GCNN,

$$\mathcal{N}^{(\ell+1)}(f) = \frac{1}{\text{vol}(G)} \int_G \mathrm{d}g \; [\mathcal{N}^{(\ell)}(f)](g) \,. \tag{10}$$

Using these layers, a wide variety of equivariant- and invariant networks with respect to a general symmetry group $G$ can be easily constructed.

## 4. Equivariant Neural Tangent Kernels

This section presents our recursive relations for the NTK and the NNGP for group convolutional layers. These recursions allow for efficient calculation of these kernels for arbitrary group convolutional architectures and thus provide the necessary tools to analytically study their training dynamics in the large width limit. Specifically, we derive recursion relations for group convolutions (4), lifting layers (6) and group pooling layers (10) by evaluating the derivatives and expectation values in (2).

### 4.1. Equivariant NTK for Group Convolutions

Since the domain of the feature maps in GCNNs is the symmetry group $G$, the layer-$\ell$ NNGP and NTK kernels do not only depend on the input feature maps $f$ and $f'$ but also on the group elements $g$, $g'$ at which the feature maps are evaluated, i.e.,

$$K_{g,g'}^{(\ell)}(f, f') = \mathbb{E}\left[ [\mathcal{N}^{(\ell)}(f)](g) \left( [\mathcal{N}^{(\ell)}(f')](g') \right)^\top \right] \,,$$

$$\Theta_{g,g'}^{(\ell)}(f, f') = \mathbb{E}\left[ \sum_{\ell'=1}^{\ell} \frac{\partial [\mathcal{N}^{(\ell)}(f)](g)}{\partial \theta^{(\ell')}} \left( \frac{\partial [\mathcal{N}^{(\ell)}(f')](g')}{\partial \theta^{(\ell')}} \right)^\top \right] \,.$$

For these kernels, we derive the following recursion relation:

**Theorem 4.1** (Kernel recursions for group convolutional layers)**.** *The layer-wise recursive relations for the NNGP and NTK of the group convolutional layer* (4) *are given by*

$$K_{g,g'}^{(\ell+1)}(f, f') = \frac{1}{|S_\kappa|} \int_{S_\kappa} \mathrm{d}h \; K_{gh,g'h}^{(\ell)}(f, f') \tag{11}$$

$$\Theta_{g,g'}^{(\ell+1)}(f, f') = K_{g,g'}^{(\ell+1)}(f, f') + \frac{1}{|S_\kappa|} \int_{S_\kappa} \mathrm{d}h \; \Theta_{gh,g'h}^{(\ell)}(f, f') \,. \tag{12}$$

*Proof.* See Appendix B.

Given $G$-invariant filter supports $S_\kappa$, these recursive definitions imply an invariance of the kernels in their group-indices under right-multiplication by the same group element $h \in G$,

$$K_{gh,g'h}^{(\ell+1)}(f, f') = K_{g,g'}^{(\ell+1)}(f, f') \tag{13}$$

$$\Theta_{gh,g'h}^{(\ell+1)}(f, f') = \Theta_{g,g'}^{(\ell+1)}(f, f') \,. \tag{14}$$

While the kernels of feature maps on the group carry $g, g'$-indices, the kernels of the input features carry $x, x'$-indices,

$$K_{x,x'}^{(0)}(f, f') = f(x)f'(x') \,, \qquad \Theta_{x,x'}^{(0)}(f, f') = 0 \,. \tag{15}$$

Using this, we also derive the following recursion relations:

**Theorem 4.2** (Kernel recursions for the lifting layer). *The layer-wise recursive relations for the NNGP and NTK of the lifting layer* (6) *are given by*[1]

$$K_{g,g'}^{(\ell+1)}(f,f') = \frac{1}{|S_\kappa|} \int_{S_\kappa} \mathrm{d}x \, K_{\rho(g)x,\,\rho(g')x}^{(\ell)}(f,f'), \quad (16)$$

$$\Theta_{g,g'}^{(\ell+1)}(f,f') = \frac{1}{|S_\kappa|} \int_{S_\kappa} \mathrm{d}x \, \Theta_{\rho(g)x,\,\rho(g')x}^{(\ell)}(f,f')$$
$$+ K_{g,g'}^{(\ell+1)}(f,f'), \quad (17)$$

*where the regular representation $\rho_{\mathrm{reg}}$ is defined in* (5).
*Proof.* See Appendix B.

The group pooling layer (10) maps feature maps on $G$ onto channel-vectors. Therefore, the kernels lose their $g, g'$-indices in this layer, as is reflected in the following result:

**Theorem 4.3** (Kernel recursions for group pooling layer). *The layer-wise recursive relations for the NNGP and NTK of the group pooling layer* (10) *are given by*

$$K^{(\ell+1)}(f,f') = \frac{1}{(\mathrm{vol}(G))^2} \int_G \mathrm{d}g \int_G \mathrm{d}g' \, K_{g,g'}^{(\ell)}(f,f')$$
$$(18)$$

$$\Theta^{(\ell+1)}(f,f') = \frac{1}{(\mathrm{vol}(G))^2} \int_G \mathrm{d}g \int_G \mathrm{d}g' \, \Theta_{g,g'}^{(\ell)}(f,f'). \quad (19)$$

*Proof.* See Appendix B.

The final layer necessary to compute kernels of GCNNs are the nonlinearities (7). Since these act pointwise on the feature maps, the recursive relations are the same as those for nonlinearities in MLPs (Jacot et al., 2018):

**Corollary 4.4** (Kernel recursions for nonlinearities). *The layer-wise recursive relations for the NNGP and NTK of the nonlinear layer* (7) *are given by*

$$\Lambda_{g,g'}^{(\ell)}(f,f') = \begin{pmatrix} K_{g,g}^{(\ell)}(f,f) & K_{g,g'}^{(\ell)}(f,f') \\ K_{g',g}^{(\ell)}(f',f) & K_{g',g'}^{(\ell)}(f',f') \end{pmatrix} \quad (20)$$

$$K_{g,g'}^{(\ell+1)}(f,f') = \mathbb{E}_{(u,v)\sim\mathcal{N}(0,\Lambda_{g,g'}^{(\ell)}(f,f'))}[\sigma(u)\sigma(v)] \quad (21)$$

$$\dot{K}_{g,g'}^{(\ell+1)}(f,f') = \mathbb{E}_{(u,v)\sim\mathcal{N}(0,\Lambda_{g,g'}^{(\ell)}(f,f'))}[\sigma'(u)\sigma'(v)] \quad (22)$$

$$\Theta_{g,g'}^{(\ell+1)}(f,f') = \dot{K}_{g,g'}^{(\ell+1)}(f,f')\Theta_{g,g'}^{(\ell)}(f,f'). \quad (23)$$

Using these results, the NTK and NNGP can be straightforwardly computed for any GCNN architecture. In particular, consider the transformation of the kernels under transformations of the inputs, i.e. consider $K(\rho_{\mathrm{reg}}(h)f, \rho_{\mathrm{reg}}(h')f')$ and $\Theta(\rho_{\mathrm{reg}}(h)f, \rho_{\mathrm{reg}}(h')f')$. From the recursion relations

[1]In practice, the lifting layer is usually the first layer, thus $\ell = 0$.

for the lifting layer in Theorem 4.2, we have the transformation property

$$K_{g,g'}^{(1)}(\rho_{\mathrm{reg}}(h)f, \rho_{\mathrm{reg}}(h')f') = K_{h^{-1}g,h'^{-1}g'}^{(1)}(f,f') \quad (24)$$

$$\Theta_{g,g'}^{(1)}(\rho_{\mathrm{reg}}(h)f, \rho_{\mathrm{reg}}(h')f') = \Theta_{h^{-1}g,h'^{-1}g'}^{(1)}(f,f'). \quad (25)$$

This left-multiplication is preserved by the recursions of both the group convolutions in Theorem 4.1 and the non-linearities in Corollary 4.4. Therefore, before any pooling layer, we have

$$K_{g,g'}^{(\ell)}(\rho_{\mathrm{reg}}(h)f, \rho_{\mathrm{reg}}(h')f') = K_{h^{-1}g,h'^{-1}g'}^{(\ell)}(f,f') \quad (26)$$

$$\Theta_{g,g'}^{(\ell)}(\rho_{\mathrm{reg}}(h)f, \rho_{\mathrm{reg}}(h')f') = \Theta_{h^{-1}g,h'^{-1}g'}^{(\ell)}(f,f'), \quad (27)$$

reflecting the equivariance of the network. The recursions of the group-pooling layer in Theorem 4.3 average over the group and the kernels become invariant after the group pooling layer

$$K^{(\ell)}(\rho_{\mathrm{reg}}(h)f, \rho_{\mathrm{reg}}(h')f') = K^{(\ell)}(f,f') \quad (28)$$

$$\Theta^{(\ell)}(\rho_{\mathrm{reg}}(h)f, \rho_{\mathrm{reg}}(h')f') = \Theta^{(\ell)}(f,f'), \quad (29)$$

as expected from an invariant network. Note that these transformation properties of the kernels are independent for both arguments.

### 4.2. Roto-Translations in the Plane

The kernel recursions provided in the previous section are valid for general symmetry groups $G$. In this section, we will specialize these expressions to the case of roto-translations in the plane with rotations by $(360/n)^\circ$. In this case, $G = C_n \ltimes \mathbb{R}^2$ where $G$ is the semidirect product of the cyclic group $C_n$ and the translation group in two dimensions $\mathbb{R}^2$. It was shown that adding this rotational symmetry to conventional CNNs boosts performance considerably for important applications such as medical image analysis (Chidester et al., 2019; Bekkers et al., 2018; Pang et al., 2023). Due to the semidirect product nature of the symmetry group, the group convolutional layers can be written as a stack of $n$ conventional convolutions which are summed over the rotation group. Details and explicit expressions for the lifting-, group convolutional- and group pooling layers in this case can be found in Appendix C.1.

The kernel recursion of ordinary CNN-layers can be written in terms of the operator (Xiao et al., 2018)

$$[\mathcal{A}_{S_\kappa}(K)](t,t') = \frac{1}{|S_\kappa|} \int_{S_\kappa} \mathrm{d}\tilde{t} \, K(t+\tilde{t}, t'+\tilde{t}), \quad (30)$$

for which efficient implementations in terms of convolutions are available in (Novak et al., 2020). In Appendix C.2, we present explicit expressions for the NNGP and NTK recursions of roto-translation equivariant convolutions in

terms of $\mathcal{A}$, retaining the efficiency of the non-rotation-equivariant kernel computations. We provide implementations of these recursions for $n = 4$ as new layers based on the `neural-tangents` package.

### 4.3. Rotations in 3d

Spherical signals subject to rotations in 3d are a further important use case with numerous applications in quantum chemistry (Duval et al., 2023), weather prediction (Bonev et al., 2023) and 3d shape recognition (Fuchs et al., 2020). The group convolutions for the corresponding symmetry group $\mathrm{SO}(3)$ can be computed efficiently in the Fourier domain, in terms of coefficients in a steerable basis of spherical harmonics $Y_m^l$ or Wigner matrices $\mathcal{D}_{mn}^l$, respectively (Cohen et al., 2018; Cohen & Welling, 2017). Due to the continuous nature of the $\mathrm{SO}(3)$ group, comprehensive data augmentation is not feasible, thus making group convolutional networks the natural choice to incorporate such symmetries. In Appendix D.1, we provide a summary of the necessary Fourier space relations for $\mathrm{SO}(3)$-equivariant networks.

The kernel forward equations (11), (12) simplify to purely algebraic equations in terms of Fourier coefficients

$$\left[ K^{\widehat{(\ell)}(f, f')} \right]_{mn,m'n'}^{l,l'} =$$
$$\int \mathrm{d}R \int \mathrm{d}R' \, K_{R,R'}^{(\ell)}(f, f') \mathcal{D}_{mn}^l(R) \mathcal{D}_{m'n'}^{l'}(R') , \quad (31)$$

and analogously for the NTK. Note that the kernels have two group indices, thus necessitating a double Fourier transform. Detailed relations for lifting-, group convolutional- and group pooling layer in the Fourier space are provided in Appendix D.2. Again, these layers are implemented in the `neural-tangents` package and the necessary generalized FFTs are provided by the JAX-based package `s2fft` (Price & McEwen, 2024).

## 5. Data Augmentation Versus Group Convolutions at Infinite Width

The recursive relations presented in the previous sections give analytical access to the training dynamics of equivariant neural networks. In particular, they allow for a more in-depth theoretical understanding of the similarities and differences of data augmentation and manifest equivariance than previously possible.

It is known that ensembles of independently initialized neural networks trained with data augmentation yield equivariant mean predictions (Gerken & Kessel, 2024; Nordenfors & Flinth, 2024). It is however unclear how these equivariant functions relate to trained manifestly equivariant networks. Using the recursive relations from Section 4.1, it is possible to show that non-equivariant networks trained with data aug-

mentation in fact converge to group convolutional networks in the ensemble mean.

### 5.1. Data Augmentation at Infinite Width

In the infinite width limit, the training dynamics under gradient descent can be solved exactly (Jacot et al., 2018). This enables us to explicitly study data augmentation, showing that data augmentation and kernel averaging yield the same mean predictions, as detailed in the following

**Theorem 5.1.** *Let $\mu_t^{\mathrm{aug}}$ and $\mu_t$ be the mean predictions after $t$ training steps of infinite ensembles of two neural network architectures $\mathcal{N}^{\mathrm{aug}}$ and $\mathcal{N}$. Let $\mathcal{N}^{\mathrm{aug}}$ be trained on the fully $G$-augmented training data of $\mathcal{N}$ and assume that the NTKs of the two architectures are related by*

$$\Theta(f, f') = \frac{1}{|G|} \sum_{g \in G} \Theta^{\mathrm{aug}}(f, \rho_{\mathrm{reg}}(g)f') . \quad (32)$$

*Then, $\mu_t^{\mathrm{aug}}$ and $\mu_t$ converge in the infinite width limit to the same function for all $t$ for quadratic losses, up to quadratic corrections in the learning rate.*

*Proof.* See Appendix E.

The proof of Theorem 5.1 proceeds inductively over training steps. At initialization, both mean functions are identically zero (Neal, 1996; Lee et al., 2018). The updates for the two networks can be written in terms of the NTK and shown to agree by splitting the sum over augmented training data into a sum over samples and a sum over $G$ and using the assumption (32).

In fact, the same argument can be used to show that the individual networks $\mathcal{N}^{\mathrm{aug}}$ and $\mathcal{N}$ (as opposed to their ensemble averages) agree if their empirical NTKs satisfy (32) as long as the networks are identical for all inputs and equivariant at initialization. This would for instance be the case if $\mathcal{N}(x) = \mathcal{N}^{\mathrm{aug}}(x) = 0$ for all $x$ at initialization.

### 5.2. Kernel Averaging Yields GCNN-Kernels

Theorem 5.1 shows equivalence of augmented and non-augmented networks if the NTKs of both architectures are related by group-averaging. Consider the case of training an MLP on augmented data. Then, (32) prompts us to consider the group-average of its NTK to find the architecture which results in the same mean predictions if trained without data augmentation. By iterating the recursive kernel-relations found in the previous section, one can in fact show that this architecture is a GCNN, as detailed in the following

**Theorem 5.2.** *Let $\mathcal{N}^{\mathrm{FC}}$ be an MLP acting on feature maps with output in $\mathbb{R}$ and architecture*

$$\mathcal{N}^{\mathrm{FC}} = \mathrm{FC}^{(L)} \circ \sigma \circ \cdots \circ \mathrm{FC}^{(3)} \circ \sigma \circ \mathrm{FC}^{(1)} , \quad (33)$$

*where FC denotes a dense MLP layer and $\sigma$ a point-wise nonlinearity. Let $\mathcal{N}^{\mathrm{GC}}$ be a G-invariant GCNN with architecture*

$$\mathcal{N}^{\mathrm{GC}} = \mathrm{GPool} \circ \mathrm{GConv}(S_\kappa^L) \circ \sigma \circ \mathrm{GConv}(S_\kappa^{L-2}) \circ \sigma \cdots$$
$$\cdots \circ \mathrm{GConv}(S_\kappa^3) \circ \sigma \circ \mathrm{Lifting}(S_\kappa^1) , \quad (34)$$

*where $S_\kappa^\ell$ are the supports of the convolutional filters with $S_\kappa^1 = X$, the domain of the input feature maps, and the other $S_\kappa^\ell$ are invariant under G. Then, the G-averages of the kernels of the MLP are given by the kernels of the GCNN,*

$$K^{\mathrm{GC}}(f, f') = \frac{1}{\mathrm{vol}(G)} \int \mathrm{d}g \ K^{\mathrm{FC}}(f, \rho_{\mathrm{reg}}(g)f') \quad (35)$$

$$\Theta^{\mathrm{GC}}(f, f') = \frac{1}{\mathrm{vol}(G)} \int \mathrm{d}g \ \Theta^{\mathrm{FC}}(f, \rho_{\mathrm{reg}}(g)f') . \quad (36)$$

*Proof.* See Appendix E.

Together with Theorem 5.1, this theorem shows that by augmenting an arbitrary deep MLP at infinite width, one obtains a specific equivariant architecture, namely a GCNN with the same depth $L$ and an additional group-pooling layer. This result singles out group convolutional layers among other equivariant layers and mirrors the fact that group convolutions are the unique linear equivariant layers under the regular representation. Note that according to Theorem 5.1 the equivalence between augmented and equivariant networks holds throughout training and even out of distribution.

### 5.3. Augmenting a CNN

Consider a generalization of the roto-translation symmetry discussed in Section 4.2, namely a general semidirect product group, $G = K \ltimes N$ with $N$ a normal subgroup of $G$. For $N$ a translation group, this covers cases such as CNNs in two and three dimensions with additional rotation or reflection symmetry (Cesa et al., 2022).

The semidirect product structure of $G$ allows a splitting of the full equivariance, namely training a $K \ltimes N$-invariant GCNN is equivalent to training an $N$-invariant GCNN on $K$-augmented data. In order to see this, we show the corresponding kernel averages for Theorem 5.1 to hold:

**Theorem 5.3.** *Let $\mathcal{N}^{K \ltimes N}$ be the $K \ltimes N$-invariant GCNN with architecture (34) and $K$-invariant filter supports $S_\kappa^\ell$ which for the GConv-layers decompose as $S_\kappa^\ell = K_\kappa^\ell \times N_\kappa^\ell$, $K_\kappa^\ell \subseteq K$, $N_\kappa^\ell \subseteq N$. Let $\mathcal{N}^N$ be the $N$-invariant GCNN with architecture (34) and filter supports $N_\kappa^L, \ldots, N_\kappa^3$ and $S_\kappa^1$. Then, the NNGPs and NTKs of these networks are*

*related by*

$$K^{K \ltimes N}(f, f') = \frac{1}{\mathrm{vol}(K)} \int_K \mathrm{d}k \ K^N(f, \rho_{\mathrm{reg}}(k)f') \quad (37)$$

$$\Theta^{K \ltimes N}(f, f') = \frac{1}{\mathrm{vol}(K)} \int_K \mathrm{d}k \ \Theta^N(f, \rho_{\mathrm{reg}}(k)f') . \quad (38)$$

*Proof.* See Appendix E.

*Remark* 5.4. For $K = C_n$ and $N = \mathbb{R}^2$, i.e. roto-translations in the plane, $\mathcal{N}^N$ becomes an ordinary CNN and $\mathcal{N}^{K \ltimes N}$ is of the form discussed in Section 4.2. According to Theorem 5.3, the kernels of the rotation-equivariant network are then given by averaging the kernels of the CNN,

$$K^{C_n \ltimes \mathbb{R}^2}(f, f') = \frac{1}{n} \sum_{r \in C_n} K^{\mathrm{CNN}}(f, \rho_{\mathrm{reg}}(r)f') \quad (39)$$

$$\Theta^{C_n \ltimes \mathbb{R}^2}(f, f') = \frac{1}{n} \sum_{r \in C_n} \Theta^{\mathrm{CNN}}(f, \rho_{\mathrm{reg}}(r)f') \quad (40)$$

if the spatial filter shapes agree for both networks and are rotation-invariant.

Taken together with Theorem 5.1, this shows that training an $N$-invariant GCNN on $K$-augmented data results in a $K \ltimes N$-invariant GCNN. For the special case of $N$ a translation group and $K$ a rotation group, this means that training a CNN on rotation-augmented data is equivalent to training a roto-translation equivariant GCNN on unaugmented data. In Section 6, we will show that this still holds approximately for finite-width networks and ensembles.

### 5.4. Distribution of Ensemble Members

Consider training two ensembles of networks with (a) data augmentation on a non-equivariant architecture and (b) no data augmentation on an equivariant architecture. Then, the distributions of the individual networks in these ensembles do not agree since most of the augmented networks will not be equivariant. However, our results show that the ensemble mean of (a) converges to the ensemble mean of (b) with a specific GCNN architecture. This establishes a highly non-trivial relation between data augmentation and GCNNs.

## 6. Experiments

In the following, we validate the theoretical results of the preceding sections experimentally for various datasets (Cifar10, QM9, MNIST, and histological data), tasks (regression and classification), and groups (SO(3) and $C_4 \ltimes \mathbb{R}^2$).

**Kernel Convergence for $C_4 \ltimes \mathbb{R}^2$.** Figure 1 confirms that the Monte-Carlo estimate of the NTK converges to our analytical expression as the width increases. Our MC estimates are obtained by replacing the expectation values in (1)

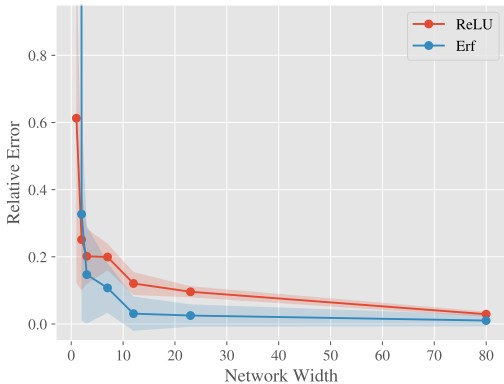

*Figure 1.* **Convergence of the Monte-Carlo estimates of the NTK to their infinite-width limits for** $G = C_4 \ltimes \mathbb{R}^2$**.** Plotted is the relative error averaged over the components of a $3 \times 3$ Gram matrix for networks with a ReLU or an error function nonlinearity. Bands show $\pm$ one standard deviation of the estimator.

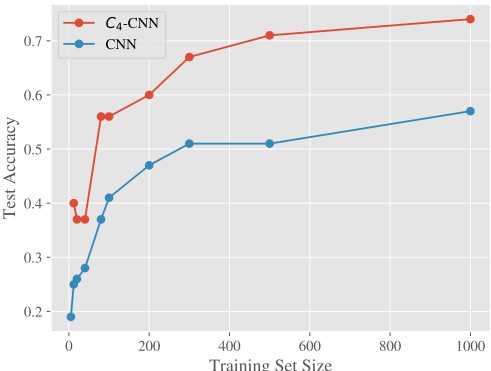

*Figure 2.* **NTK for image classification.** Test accuracy of the arising NTK kernel methods in the infinite width and infinite training time limit for different training set sizes. The results for both a conventional CNN and a $C_4 \ltimes \mathbb{R}^2$-invariant GCNN are shown.

by the sample mean of 1000 initializations of the network. We considered GCNNs with one lifting- and four group-convolution layers interspersed with ReLU nonlinearites, followed by a group-pooling layer. The convergence of the NNGP is shown in a similar plot in Appendix F.1.

**Medical Image Classification with** $C_4 \ltimes \mathbb{R}^2$**.** We show that rotation-equivariant NTK-predictors outperform non-equivariant NTK-predictors on a dataset of histological images (Kather et al., 2018) containing nine distinct classes of tissues. Specifically, we compare a CNN architecture with the corresponding rotation-invariant GCNN architecture, in which we replace each of the five convolutional layers with a group-convolutional- or lifting layer, respectively and used a group-pooling layer instead of a `SumPool` layer. Figure 2 shows the improved scaling behavior of the equivariant kernel with training set size upon using the infinite-time

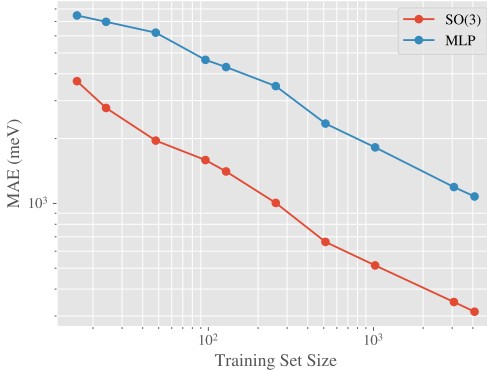

*Figure 3.* **NTK for molecular energy prediction.** Molecular energy MAEs of the NTK kernel methods in the infinite width and training time limit for different training set sizes. The results are for both a conventional MLP and a $SO(3)$-invariant GCNN.

solution of the NTK-dynamics under MSE loss,

$$\mu(x) = \Theta(x, \mathcal{X})\Theta(\mathcal{X}, \mathcal{X})^{-1}\mathcal{Y}, \tag{41}$$

where $\mathcal{X}$ represents the training images, scaled down to $32 \times 32$ pixels, $\mathcal{Y}$ are the training labels, given by $e_c - \frac{1}{9}\mathbf{1}$ for class $c$, and $\Theta(\mathcal{X}, \mathcal{X})$ is the Gram matrix of the NTK. We refer to Appendix F.2 for more details.

**Molecular Energy Regression with** $SO(3)$**.** We benchmark the NTK-predictor resulting from an $SO(3)$-invariant network on the *QM9* dataset (Ramakrishnan et al., 2014) by predicting molecular energies $U_0$ from atom configurations utilizing (41). Comparing this to the corresponding MLP kernel, we observe a considerable performance boost for the invariant kernel over a range of training set sizes, as shown in Figure 3. For preprocessing, we construct spherical signals from the atom configurations as described in (Esteves et al., 2023). For each atom $i$ of the at most 29 atoms, the environment is represented by pairwise Gaussian smearing over atoms with the same atomic number $z$

$$f_{i,z,p}(x) = \sum_{j:z_j=z} \frac{z_i z}{\|r_{ij}\|^p} e^{-\frac{1}{\beta}\left(\frac{r_{ij}}{\|r_{ij}\|}\cdot x - 1\right)^2}. \tag{42}$$

Choosing $p \in \{2, 6\}$ and considering all of QM9's five atom types leads to 29 spherical per-atom signals with $5 \times 2$ channels each. Each of those per-atom signals are then either processed by a two layer $SO(3)$-equivariant network with group pooling on top or by a two-layer MLP. The per-atom outputs are eventually summed and fed into a final fully-connected layer similarly as in (Cohen et al., 2018). The input signals are constructed at a resolution of $12 \times 11$ on the sphere, corresponding to a bandlimit of $L = 6$, which are then downsampled to a bandlimit $L = 3$ for the group layer. We provide further details in Appendix F.3.

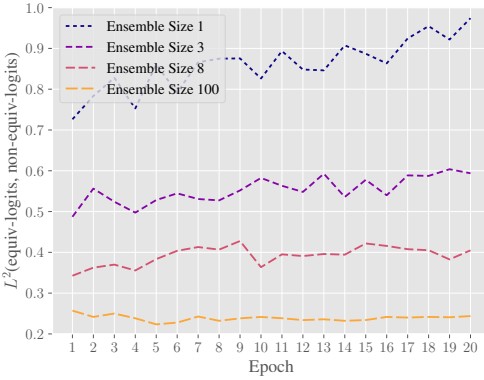

Figure 4. **Convergence of finite-width ensembles trained with data augmentation to ensembles of GCNNs on MNIST.** $L^2$-distance between the logits of the equivariant- and non-equivariant ensemble trained with data augmentation for different ensemble sizes on out of distribution data. For larger ensembles, the distance decreases.

**Data Augmentation Versus Group Convolutions at Finite Width.** In Section 5, we proved that networks trained with data augmentation converge in expectation to group convolutional networks at infinite width. We also verify empirically that the mean predictions agree progressively with increasing ensembles size at finite width, supporting approximate validity away from the theoretical limit. To this end, we train ensembles of CNNs and GCNNs with symmetry group $C_4 \ltimes \mathbb{R}^2$ as discussed in Remark 5.4 on CIFAR10 and MNIST using the MSE-loss against smoothed one-hot labels as for the medical images above. For implementing the GCNNs, we used the `escnn`-package (Weiler & Cesa, 2019). As shown in Figure 4 for MNIST, the outputs of both ensembles converge to the same vector for large ensemble sizes throughout training and even out of distribution. For further details on the model architectures, out of distribution data, as well as results on CIFAR10 and histological images, see Appendix F.4.

## 7. Conclusion

This paper provides recursive relations for the NNGP and NTK for group convolutional neural networks allowing us to theoretically establish an interesting equivalence between equivariance-based to data-augmentation-based training dynamics. We also show that equivariant kernels outperform their non-equivariant counterparts as kernel machines.

A careful comparison of equivariant GCNNs and equivariant data augmentation, beyond the invariant case analyzed in Section 5, would be an interesting subject for further research. In particular, Theorem 5.1 can be straightforwardly extended to the equivariant case, as demonstrated in Appendix E. However, Theorems 5.2 and 5.3 rely on the group

pooling layer and are thus specialized to invariant GCNNs. An equivariant extension would require new layers beyond those presented here since in the infinite-width limit, the NTK of an MLP becomes proportional to the unit matrix in the output channels, trivializing the feature map.

## Acknowledgments

J.G. and P.M. want to thank Max Guillen for inspiring discussions and collaborations on related projects. The work of J.G. and P.M. is supported by the Wallenberg AI, Autonomous Systems and Software Program (WASP) funded by the Knut and Alice Wallenberg Foundation. The computations were enabled by resources provided by the National Academic Infrastructure for Supercomputing in Sweden (NAISS) and the Swedish National Infrastructure for Computing (SNIC) at C3SE partially funded by the Swedish Research Council through grant agreements 2022-06725 and 2018-05973.

## Impact Statement

This paper presents work whose goal is to advance the field of Machine Learning. There are many potential societal consequences of our work, none which we feel must be specifically highlighted here.

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

## A. Basics of Neural Tangent Kernel Theory

A neural network $\mathcal{N} : \mathbb{R}^{n_{\text{in}}} \to \mathbb{R}^{n_{\text{out}}}$ which is trained using continuous gradient descent

$$\frac{\mathrm{d}\theta}{\mathrm{d}t} = -\eta \frac{\partial \mathcal{L}}{\partial \theta} \tag{43}$$

on a loss function $\mathcal{L}$ with learning rate $\eta$ evolves according to

$$\frac{\mathrm{d}\mathcal{N}(x)}{\mathrm{d}t} = -\eta \sum_{i=1}^{n_{\text{train}}} \Theta_t(x, x_i) \frac{\partial \mathcal{L}}{\partial \mathcal{N}(x_i)} \,, \tag{44}$$

where the sum runs over the training samples $x_i$ and $\Theta_t \in \mathbb{R}^{n_{\text{out}} \times n_{\text{out}}}$ is the NTK

$$\Theta_t(x, x') = \frac{\partial \mathcal{N}(x)}{\partial \theta} \left( \frac{\partial \mathcal{N}(x')}{\partial \theta} \right)^\top . \tag{45}$$

For finite-width networks, $\Theta_t$ depends on the initialization and the training time and is referred to as the *empirical* NTK. At infinite width, $\Theta_t$ becomes independent of the initialization (it still depends on the initialization distribution) since it approaches its expectation value over initializations

$$\Theta(x, x') = \mathbb{E}_{\theta \sim p_{\text{init}}} \left[ \frac{\partial \mathcal{N}(x)}{\partial \theta} \left( \frac{\partial \mathcal{N}(x')}{\partial \theta} \right)^\top \right] . \tag{46}$$

It furthermore becomes constant throughout training and proportional to the unit matrix (Jacot et al., 2018) in the NTK parametrization. For this reason, we drop the $t$-subscript on this *frozen* NTK and treat it as a scalar. In the following, we will always mean (46) when we refer to the NTK unless otherwise stated. The NTK parametrization of a linear layer has an additional $1/\sqrt{n_{\text{fan in}}}$ prefactor and uses independent standard Gaussians as initialization distributions. Hence, an MLP layer is given by

$$\mathcal{N}^{(\ell)}(x) = \sigma\big(\mathcal{N}^{(\ell-1)}(x)\big) = \sigma\left( \frac{1}{\sqrt{n_{\text{fan in}}}} W \mathcal{N}^{(\ell-2)}(x) + b \right) , \tag{47}$$

with nonlinearity $\sigma$, weights $W$ and bias $b$.

## B. Proofs: Kernel Recursions for GCNN-Layers

In this section, we provide proofs for the theorems given in Section 4.1 in the main text.

**Theorem 4.1** (Kernel recursions for group convolutional layers). *The layer-wise recursive relations for the NNGP and NTK of the group convolutional layer* (4) *are given by*

$$K_{g,g'}^{(\ell+1)}(f, f') = \frac{1}{|S_\kappa|} \int_{S_\kappa} \mathrm{d}h \ K_{gh,g'h}^{(\ell)}(f, f') \tag{11}$$

$$\Theta_{g,g'}^{(\ell+1)}(f, f') = K_{g,g'}^{(\ell+1)}(f, f') + \frac{1}{|S_\kappa|} \int_{S_\kappa} \mathrm{d}h \, \Theta_{gh,g'h}^{(\ell)}(f, f') \,. \tag{12}$$

*Proof.* We first compute the NNGP recursion relation. For group-convolution layers, the definition (3) of the NNGP reads

$$K_{g,g'}^{(\ell+1)}(f, f') = \mathbb{E}\left[ [\mathcal{N}^{(\ell+1)}(f)](g) \, [\mathcal{N}^{(\ell+1)}(f')](g') \right] \tag{48}$$

$$= \frac{1}{S_\kappa} \int_G \mathrm{d}h \, \mathrm{d}h' \ \mathbb{E}\left[ \kappa^{(\ell+1)}(g^{-1}h) \kappa^{(\ell+1)}(g'^{-1}h') \right] \mathbb{E}\left[ [\mathcal{N}^{(\ell)}(f)](h) [\mathcal{N}^{(\ell)}(f')](h') \right] , \tag{49}$$

where we have again dropped the $1/\sqrt{n_\ell}$-prefactors and channel dependencies since these converge to the expectation value in the infinite width limit. Next, we shift the integration variables by $g$ and $g'$ which leaves the Haar measure invariant by its definition

$$K_{g,g'}^{(\ell+1)}(f, f') = \frac{1}{S_\kappa} \int_G \mathrm{d}h \, \mathrm{d}h' \ \mathbb{E}\left[ \kappa^{(\ell+1)}(h) \kappa^{(\ell+1)}(h') \right] \mathbb{E}\left[ [\mathcal{N}^{(\ell)}(f)](gh) [\mathcal{N}^{(\ell)}(f')](g'h') \right] . \tag{50}$$

Since the kernel components are sampled independently from standard Gaussians at initialization, we only obtain a contribution to the integral when $h = h'$ and $\kappa^{(\ell+1)}$ has support at this point, i.e.

$$K_{g,g'}^{(\ell+1)}(f, f') = \frac{1}{S_\kappa} \int_{S_\kappa} \mathrm{d}h \; \mathbb{E}\left[ [\mathcal{N}^{(\ell)}(f)](gh)[\mathcal{N}^{(\ell)}(f')](g'h) \right]. \tag{51}$$

Comparing to (3) shows that the right-hand side is just the NNGP $K^{(\ell)}(f, f')$ of the previous layer evaluated at group indices $gh$ and $g'h$. This proves the NNGP recursion relation stated in the theorem.

For the NTK recursion relation, we start by specializing the general expression (2) to group-convolution layers and adapting it to the functional framework used for feature maps

$$\begin{aligned}
&\Theta_{g,g'}^{(\ell+1)}(f, f') \\
&= \int_{S_\kappa} \mathrm{d}h \; \mathbb{E}\left[ \frac{\delta[\mathcal{N}^{(\ell+1)}(f)](g)}{\delta\kappa^{(\ell+1)}(h)} \frac{\delta[\mathcal{N}^{(\ell+1)}(f')](g')}{\delta\kappa^{(\ell+1)}(h)} \right] \\
&+ \int_G \mathrm{d}\tilde{g} \, \mathrm{d}\tilde{g}' \; \mathbb{E}\left[ \frac{\delta[\mathcal{N}^{(\ell+1)}(f)](g)}{\delta[\mathcal{N}^{(\ell)}(f)](\tilde{g})} \underbrace{\left( \sum_{\ell'=1}^{\ell} \int_{S_\kappa} \mathrm{d}\tilde{h} \; \frac{\delta[\mathcal{N}^{(\ell)}(f)](\tilde{g})}{\delta\kappa^{(\ell')}(\tilde{h})} \frac{\delta\mathcal{N}^{(\ell)}(\tilde{g}')}{\delta\kappa^{(\ell')}(\tilde{h})} \right)}_{\Theta_{\tilde{g},\tilde{g}'}^{(\ell)}(f,f')} \frac{\delta[\mathcal{N}^{(\ell+1)}(f')](g')}{\delta[\mathcal{N}^{(\ell)}(f')](\tilde{g}')} \right]
\end{aligned} \tag{52}$$

According to the layer definition (4), the derivatives evaluate to

$$\frac{\delta[\mathcal{N}^{(\ell+1)}(f)](g)}{\delta\kappa^{(\ell+1)}(h)} = \frac{1}{\sqrt{n_\ell S_\kappa}} [\mathcal{N}^{(\ell)}(f)](gh) \tag{53}$$

$$\frac{\delta[\mathcal{N}^{(\ell+1)}(f)](g)}{\delta[\mathcal{N}^{(\ell)}(f)](\tilde{g})} = \frac{1}{\sqrt{n_\ell S_\kappa}} \kappa^{(\ell+1)}(g^{-1}\tilde{g}). \tag{54}$$

Therefore, (52) becomes

$$\begin{aligned}
\Theta_{g,g'}^{(\ell+1)}(f, f') &= \frac{1}{S_\kappa} \int_{S_\kappa} \mathrm{d}h \; \mathbb{E}\left[ [\mathcal{N}^{(\ell)}(f)](gh)[\mathcal{N}^{(\ell)}(f')](g'h) \right] \\
&+ \frac{1}{S_\kappa} \int_G \mathrm{d}\tilde{g} \, \mathrm{d}\tilde{g}' \; \Theta_{\tilde{g},\tilde{g}'}^{(\ell)}(f, f') \, \mathbb{E}\left[ \kappa^{(\ell+1)}(g^{-1}\tilde{g}) \kappa^{(\ell+1)}(g'^{-1}\tilde{g}') \right]
\end{aligned} \tag{55}$$

$$= K_{g,g'}^{(\ell+1)}(f, f') + \frac{1}{S_\kappa} \int_{S_\kappa} \mathrm{d}h \; \Theta_{gh,g'h}^{(\ell)}(f, f'), \tag{56}$$

where we have dropped the channel-prefactors as usual. The last line is just the NTK recursion to be proven. $\square$

**Theorem 4.2** (Kernel recursions for the lifting layer). *The layer-wise recursive relations for the NNGP and NTK of the lifting layer (6) are given by[2]*

$$K_{g,g'}^{(\ell+1)}(f, f') = \frac{1}{|S_\kappa|} \int_{S_\kappa} \mathrm{d}x \; K_{\rho(g)x, \, \rho(g')x}^{(\ell)}(f, f'), \tag{16}$$

$$\begin{aligned}
\Theta_{g,g'}^{(\ell+1)}(f, f') &= \frac{1}{|S_\kappa|} \int_{S_\kappa} \mathrm{d}x \; \Theta_{\rho(g)x, \, \rho(g')x}^{(\ell)}(f, f') \\
&+ K_{g,g'}^{(\ell+1)}(f, f'),
\end{aligned} \tag{17}$$

*where the regular representation $\rho_{\mathrm{reg}}$ is defined in (5).*

---

[2]In practice, the lifting layer is usually the first layer, thus $\ell = 0$.

*Proof.* The NNGP of the lifting layer (6) is given by

$$K_{g,g'}^{(\ell+1)}(f, f') = \mathbb{E}\left[[\mathcal{N}^{(\ell+1)}(f)](g)\,[\mathcal{N}^{(\ell+1)}(f')](g')\right] \tag{57}$$

$$= \frac{1}{S_\kappa} \int_X \mathrm{d}x\,\mathrm{d}x'\,\mathbb{E}\left[\kappa(\rho(g^{-1})x)\kappa(\rho(g'^{-1})x')\right]\mathbb{E}\left[[\mathcal{N}^{(\ell)}(f)](x)[\mathcal{N}^{(\ell)}(f')](x')\right] \tag{58}$$

$$= \frac{1}{S_\kappa} \int_X \mathrm{d}x\,\mathrm{d}x'\,\mathbb{E}\left[\kappa(x)\kappa(x')\right]\mathbb{E}\left[[\mathcal{N}^{(\ell)}(f)](\rho(g)x)[\mathcal{N}^{(\ell)}(f')](\rho(g')x')\right] \tag{59}$$

$$= \frac{1}{S_\kappa} \int_{S_\kappa} \mathrm{d}x\,\mathbb{E}\left[[\mathcal{N}^{(\ell)}(f)](\rho(g)x)[\mathcal{N}^{(\ell)}(f')](\rho(g')x)\right] \tag{60}$$

$$= \frac{1}{S_\kappa} \int_{S_\kappa} \mathrm{d}x\, K_{\rho(g)x,\rho(g')x}^{(\ell)}(f, f')\,, \tag{61}$$

where we have moved the regular representation through $\mathcal{N}^{(\ell)}$ onto $f$ by using equivariance. This proves the NNGP recursion-relation.

According to (2), the NTK recursion evaluates to

$$\Theta_{g,g'}^{(\ell+1)}(f, f') = \int_{S_\kappa} \mathrm{d}x\,\mathbb{E}\left[\frac{\delta[\mathcal{N}^{(\ell+1)}(f)](g)}{\delta\kappa^{(\ell+1)}(x)}\frac{\delta[\mathcal{N}^{(\ell+1)}(f')](g')}{\delta\kappa^{(\ell+1)}(x)}\right]$$
$$+ \int_X \mathrm{d}\tilde{x}\,\mathrm{d}\tilde{x}'\,\mathbb{E}\left[\frac{\delta[\mathcal{N}^{(\ell+1)}(f)](g)}{\delta[\mathcal{N}^{(\ell)}(f)](\tilde{x})}\Theta_{\tilde{x},\tilde{x}'}^{(\ell)}(f, f')\frac{\delta[\mathcal{N}^{(\ell+1)}(f')](g')}{\delta[\mathcal{N}^{(\ell)}(f')](\tilde{x}')}\right]. \tag{62}$$

The derivatives in this expression are given by

$$\frac{\delta[\mathcal{N}^{(\ell+1)}(f)](g)}{\delta\kappa^{(\ell+1)}(x)} = \frac{1}{\sqrt{n_\ell S_\kappa}}[\mathcal{N}^{(\ell)}(f)](\rho(g)x) \tag{63}$$

$$\frac{\delta[\mathcal{N}^{(\ell+1)}(f)](g)}{\delta[\mathcal{N}^{(\ell)}(f)](\tilde{x})} = \frac{1}{\sqrt{n_\ell S_\kappa}}\kappa^{(\ell+1)}(\rho(g^{-1})\tilde{x})\,. \tag{64}$$

Plugging this back into (62) yields the desired NTK recursion relation,

$$\Theta_{\tilde{x},\tilde{x}'}^{(\ell)}(f, f') = \frac{1}{S_\kappa} \int_{S_\kappa} \mathrm{d}x\,\mathbb{E}\left[[\mathcal{N}^{(\ell)}(f)](\rho(g)x)[\mathcal{N}^{(\ell)}(f')](\rho(g')x)\right]$$
$$+ \frac{1}{S_\kappa} \int_X \mathrm{d}\tilde{x}\,\mathrm{d}\tilde{x}'\,\Theta_{\tilde{x},\tilde{x}'}^{(\ell)}(f, f')\mathbb{E}\left[\kappa^{(\ell+1)}(\rho(g^{-1})\tilde{x})\kappa^{(\ell+1)}(\rho(g'^{-1})\tilde{x}')\right] \tag{65}$$

$$= K_{g,g'}^{(\ell+1)}(f, f') + \frac{1}{S_\kappa} \int_{S_\kappa} \mathrm{d}x\,\Theta_{\rho(g)x,\,\rho(g')x'}^{(\ell)}(f, f')\,. \tag{66}$$

$$\square$$

**Theorem 4.3** (Kernel recursions for group pooling layer). *The layer-wise recursive relations for the NNGP and NTK of the group pooling layer* (10) *are given by*

$$K^{(\ell+1)}(f, f') = \frac{1}{(\mathrm{vol}(G))^2} \int_G \mathrm{d}g \int_G \mathrm{d}g'\, K_{g,g'}^{(\ell)}(f, f') \tag{18}$$

$$\Theta^{(\ell+1)}(f, f') = \frac{1}{(\mathrm{vol}(G))^2} \int_G \mathrm{d}g \int_G \mathrm{d}g'\, \Theta_{g,g'}^{(\ell)}(f, f')\,. \tag{19}$$

*Proof.* Since we integrate over the entire domain of the input feature maps $\mathcal{N}^{(\ell)}(f) : G \to \mathbb{R}^{n_\ell}$ in the pooling layer (10), are the output features $\mathcal{N}^{(\ell)}(f) \in \mathbb{R}^{n_{\ell+1}}$ a channel-vector. Therefore, the NNGP of the group pooling layer is given by

$$K^{(\ell+1)}(f, f') = \mathbb{E}\left[\mathcal{N}^{(\ell+1)}(f)\,\mathcal{N}^{(\ell+1)}(f')\right] \tag{67}$$

$$= \frac{1}{(\mathrm{vol}(G))^2} \int_G \mathrm{d}g \int_G \mathrm{d}g'\,\mathbb{E}\left[[\mathcal{N}^{(\ell)}(f)](g)\,[\mathcal{N}^{(\ell)}(f)](g')\right] \tag{68}$$

$$= \frac{1}{(\mathrm{vol}(G))^2} \int_G \mathrm{d}g \int_G \mathrm{d}g'\, K_{g,g'}^{(\ell)}(f, f')\,. \tag{69}$$

The NTK recursion (2) is in this case

$$\Theta^{(\ell+1)}(f, f') = \int_G \mathrm{d}g \int_G \mathrm{d}g' \, \mathbb{E}\left[ \frac{\delta \mathcal{N}^{(\ell+1)}(f)}{\delta[\mathcal{N}^{(\ell)}(f)](g)} \Theta^{(\ell)}_{g,g'}(f, f') \frac{\delta \mathcal{N}^{(\ell+1)}(f')}{\delta[\mathcal{N}^{(\ell)}(f')](g')} \right] \tag{70}$$

$$= \frac{1}{(\mathrm{vol}(G))^2} \int_G \mathrm{d}g \int_G \mathrm{d}g' \, \Theta^{(\ell)}_{g,g'}(f, f') \,, \tag{71}$$

which is the NTK-relation to be shown. $\qquad\square$

## C. Equivariant Kernels for Roto-Translations in the Plane

In this appendix, we provide explicit expressions for the kernel recursions of lifting-, group convolutional- and group pooling layers for the special case of roto-translations in the plane, i.e. for the symmetry group $G = C_n \ltimes \mathbb{R}^2$. In this case, the general expressions in Theorems 4.1, 4.2 and 4.3 simplify and can be written in terms of the $\mathcal{A}$ operator (30) which can be computed efficiently in terms of ordinary 2d convolutions. However, before discussing the kernel recursions, we will first establish a simplifying notation for the GCNN layer-definitions.

### C.1. GCNNs for $G = C_n \ltimes \mathbb{R}^2$

Due to the semidirect-product structure of $G$, any element $g \in G$ can be written uniquely as a product of a translation and a rotation, $g = tr$ with $t \in \mathbb{R}^2$ and $r \in C_n$[3]. We can therefore write a feature map $\mathcal{N}^{(\ell)}$ on $G$ as a stack of $n$ feature maps on $\mathbb{R}^2$,

$$\mathcal{N}^{(\ell)}(g = tr) = \mathcal{N}^{(\ell)}_r(t) \,. \tag{72}$$

Using this representation, the lifting- and group convolutional layers can be written in terms of ordinary two-dimensional convolutions as (Cohen & Welling, 2016)

$$[\mathcal{N}^{(1)}(f)]_r(t) = \frac{1}{\sqrt{n_{\mathrm{in}}|S_\kappa|}} \int_{\mathbb{R}^2} \mathrm{d}x \, \kappa\big(\rho(r^{-1})(x - t)\big) f(x) \tag{73}$$

$$[\mathcal{N}^{(\ell+1)}(f)]_r(t) = \frac{1}{\sqrt{n_\ell |S_\kappa|}} \sum_{r' \in C_n} \int_{\mathbb{R}^2} \mathrm{d}t' \, \kappa_{r^{-1}r'}\big(\rho(r^{-1})(t' - t)\big) [\mathcal{N}^{(\ell)}(f)]_{r'}(t') \,, \quad \ell \geq 1 \,, \tag{74}$$

where $\rho$ is the fundamental representation of $\mathrm{SO}(2)$ on $\mathbb{R}^2$, given by two-dimensional rotation matrices.

Finally, for invariant problems like classification, the group pooling layer (10) is central to making the network invariant. For $C_n \ltimes \mathbb{R}^2$, it is given by

$$[\mathcal{N}^{(\ell+1)}(f)] = \frac{1}{\sqrt{n|\mathrm{supp}(\mathcal{N}^{(\ell)}(f))|}} \sum_{r \in C_n} \int \mathrm{d}x \, [\mathcal{N}^{(\ell)}(f)]_r(x) \,. \tag{75}$$

### C.2. Kernel recursions for $G = C_n \ltimes \mathbb{R}^2$

In analogy to the notation introduced in the previous section for feature maps, we write for the NNGP and NTK on $C_n \ltimes \mathbb{R}^2$

$$K_{g=tr, g'=t'r'}(f, f') = [K_{rr}(f, f')](t, t') \,, \qquad \Theta_{g=tr, g'=t'r'}(f, f') = [\Theta_{rr'}(f, f')](t, t') \tag{76}$$

to emphasize the dependency on the two translations $t, t' \in \mathbb{R}^2$. Furthermore, we repeat here the definition of the operator (30) for convenience

$$[\mathcal{A}_{S_\kappa}(K)](t, t') = \frac{1}{|S_\kappa|} \int_{S_\kappa} \mathrm{d}\tilde{t} \, K(t + \tilde{t}, t' + \tilde{t}) \,, \tag{77}$$

Given these definitions, the recursive relations from Theorem 4.1 for group convolutions can be computed efficiently using the following

---

[3]In an abuse of notation, we will denote both the abstract translation group element and its representation as a vector in $\mathbb{R}^2$ by the same symbol.

**Lemma C.1** (Kernel recursions of group convolutional layers for roto-translations). *In the case $G = C_n \ltimes \mathbb{R}^2$, the layer-wise recursive relations for the NNGP and NTK of the group convolutional layer* (74) *are given by*

$$[K_{rr'}^{(\ell+1)}(f, f')](t, t') = \sum_{\tilde{r} \in C_n} [\mathcal{A}_{\rho(r)S_\kappa}(\tilde{K}_{r\tilde{r}, r'\tilde{r}}^{(\ell)}(f, f'))](t, \rho(rr'^{-1})t') \tag{78}$$

$$[\Theta_{rr'}^{(\ell+1)}(f, f')](t, t') = [K_{rr'}^{(\ell+1)}(f, f')](t, t') + \sum_{\tilde{r} \in C_n} [\mathcal{A}_{\rho(r)S_\kappa}(\tilde{\Theta}_{r\tilde{r}, r'\tilde{r}}^{(\ell)}(f, f'))](t, \rho(rr'^{-1})t'), \tag{79}$$

*where*

$$[\tilde{K}_{r\,r'}^{(\ell)}(f, f')](t, t') = [K_{r\,r'}^{(\ell)}(f, f')](t, \rho(r'r^{-1})t') \tag{80}$$

$$\tilde{\Theta}_{r\,r'}^{(\ell)}(f, f') = [\Theta_{r\,r'}^{(\ell)}(f, f')](t, \rho(r'r^{-1})t'). \tag{81}$$

*Proof.* In order to compute the NNGP recursion relation in the notation (76), we first need to compute the unique decomposition of a general group multiplication $gh$, $g, h \in G$ into a rotation and a translation. This is possible since $G$ is a semidirect product group. Starting from $g = t_g r_g$ and $h = t_h r_h$ with $t_g, t_h \in \mathbb{R}^2$ and $r_g, r_h \in C_n$, we have

$$gh = t_g r_g t_h r_h = t_g r_g t_h r_g^{-1} r_g r_h. \tag{82}$$

Since $\mathbb{R}^2$ is a normal subgroup of $G$ (a further property implied by the semidirect product), $r_g t_h r_g^{-1} \in \mathbb{R}^2$. Therefore,

$$t_{gh} = t_g r_g t_h r_g^{-1} \in \mathbb{R}^2 \qquad \text{and} \qquad r_{gh} = r_g r_h \in C_n. \tag{83}$$

Since the action of $t_{gh}$ on a vector $x \in \mathbb{R}^2$ is given by

$$\rho(t_{gh})x = x + \rho(r_g^{-1})t_h + t_g, \tag{84}$$

we obtain for the NNGP recursion from Theorem 4.1,

$$[K_{rr'}^{(\ell+1)}(f, f')](t, t') = \frac{1}{|S_\kappa|} \sum_{\tilde{r} \in C_4} \int_{S_\kappa} d\tilde{t} \, [K_{r\tilde{r}, r'\tilde{r}}^{(\ell)}(f, f')](\rho(r^{-1})\tilde{t} + t, \rho(r'^{-1})\tilde{t} + t'). \tag{85}$$

This we will now write in terms of the $\mathcal{A}$-operator (77). However, since the $\mathcal{A}$-operator shifts both slots of the argument kernel by $y$, whereas the first argument in (85) is shifted by $\rho(r^{-1})\tilde{t}$, while the second argument is shifted by $\rho(r'^{-1})\tilde{t}$, we cannot write (85) directly in terms of $\mathcal{A}(K)$, but need to compute $\mathcal{A}$ at a transformed argument instead and then transform back. To this end, first consider

$$[\mathcal{A}_{S_\kappa}(\tilde{K}_{rr'}^{(\ell)}(f, f'))](t, t') = \frac{1}{|S_\kappa|} \int_{S_\kappa} d\tilde{t} \, [\tilde{K}_{rr'}^{(\ell)}(f, f')](t + \tilde{t}, t' + \tilde{t}) \tag{86}$$

$$= \frac{1}{|S_\kappa|} \int_{S_\kappa} d\tilde{t} \, [K_{rr'}^{(\ell)}(f, f')](t + \tilde{t}, \rho(r'r^{-1})(t' + \tilde{t})). \tag{87}$$

Therefore, we obtain for the RHS of the NNGP recursion

$$\sum_{\tilde{r} \in C_n} [\mathcal{A}_{\rho(r)S_\kappa}(\tilde{K}_{r\tilde{r}, r'\tilde{r}}^{(\ell)}(f, f'))](t, \rho(rr'^{-1})t')$$

$$= \frac{1}{|S_\kappa|} \sum_{\tilde{r} \in C_n} \int_{\rho(r)S_\kappa} d\tilde{t} \, [K_{r\tilde{r}, r'\tilde{r}}^{(\ell)}(f, f')](t + \tilde{t}, \rho(r'r^{-1})(\rho(rr'^{-1})t' + \tilde{t})) \tag{88}$$

$$= \frac{1}{|S_\kappa|} \sum_{\tilde{r} \in C_n} \int_{\rho(r)S_\kappa} d\tilde{t} \, [K_{r\tilde{r}, r'\tilde{r}}^{(\ell)}(f, f')](t + \tilde{t}, t' + \rho(r'r^{-1})\tilde{t}) \tag{89}$$

$$= \frac{1}{|S_\kappa|} \sum_{\tilde{r} \in C_n} \int_{S_\kappa} d\tilde{t} \, [K_{r\tilde{r}, r'\tilde{r}}^{(\ell)}(f, f')](t + \rho(r)\tilde{t}, t' + \rho(r')\tilde{t}) \tag{90}$$

The last line is just (85), proving the NNGP recursion relation.

For the NTK, we start from the NTK-recursion in Theorem 4.1. The structure of the integral appearing in that recursion is the same as the one of the integral in the NNGP recursion. Therefore, the NTK recursion is given by

$$
\begin{aligned}
[\Theta_{rr'}^{(\ell+1)}(f, f')](t, t') &= [K_{rr'}^{(\ell+1)}(f, f')](t, t') \\
&+ \frac{1}{S_\kappa} \sum_{\tilde{r} \in C_4} \int_{S_\kappa} d\tilde{t} \; [\Theta_{r\tilde{r}, r'\tilde{r}}^{(\ell)}(f, f')](\rho(r^{-1})\tilde{t} + t, \rho(r'^{-1})\tilde{t} + t').
\end{aligned}
\tag{91}
$$

The integral can be written in terms of the $\mathcal{A}$-operator following the same steps as for the NNGP above. $\qquad\square$

Therefore, by first computing the kernels $\tilde{K}^{(\ell)}$ and $\tilde{\Theta}^{(\ell)}$ and then applying the $\mathcal{A}$-operator, it is possible to efficiently compute the kernel-recursions in this case. Similarly, the recursive kernel-relations for the lifting layer can also be written efficiently in terms of the $\mathcal{A}$-operator, as detailed in

**Lemma C.2** (Kernel recursions of lifting layers for roto-translations). *The layer-wise recursive relations for the NNGP and NTK of the group convolutional layer* (73) *are given by*

$$
[K_{rr'}^{(\ell+1)}(f, f')](t, t') = \left[\mathcal{A}_{\rho(r)S_\kappa}(\tilde{K}_{r\,r'}^{(\ell)}(f, f'))\right](t, \rho(rr'^{-1})t')
\tag{92}
$$

$$
[\Theta_{rr'}^{(\ell+1)}(f, f')](t, t') = [K_{rr'}^{(\ell+1)}(f, f')](t, t') + \left[\mathcal{A}_{\rho(r)S_\kappa}(\tilde{\Theta}_{r\,r'}^{(\ell)}(f, f'))\right](t, \rho(rr'^{-1})t'),
\tag{93}
$$

*where*

$$
[\tilde{K}_{rr'}^{(\ell)}(f, f')](t, t') = [K^{(\ell)}(f, f')](t, \rho(r'r^{-1})t')
\tag{94}
$$

$$
[\tilde{\Theta}_{rr'}^{(\ell)}(f, f')](t, t') = [\Theta^{(\ell)}(f, f')](t, \rho(r'r^{-1})t').
\tag{95}
$$

*Proof.* According to Theorem 4.2, the NNGP recursion is given by

$$
[K_{rr'}^{(\ell+1)}(f, f')](t, t') = \frac{1}{S_\kappa} \int_{S_\kappa} dx \; K^{(\ell)}\left(\rho(r)x + t, \rho(r')x + t'\right).
\tag{96}
$$

Comparing this expression to (85) shows that the sum over $\tilde{r}$ as well as the $r, r'$-indices on $K^{(\ell)}$ are absent in (96) but otherwise the two expressions agree. Therefore, we can use the same argument as above to rewrite (96) in terms of the $\mathcal{A}$-operator and only need to drop the $r, r'$-indices in the definition of $\tilde{K}^{(\ell)}$ as well as pick the $\tilde{r} = e$ contribution in the sum. Similarly, we can show the NTK recursion relation starting form the NTK-recursion in Theorem 4.2. $\qquad\square$

Finally, in the group pooling layer, the kernels are trivialized over their $r$- and $t$-indices, resulting in a kernel without spatial indices:

**Lemma C.3** (Kernel recursions of group pooling layers for roto-translations). *The layer-wise recursive relations for the NNGP and NTK of the group convolutional layer* (75) *are given by*

$$
K^{(\ell+1)}(f, f') = \frac{1}{n|\operatorname{supp}(\mathcal{N}^{(\ell)}(f))|} \sum_{r, r' \in C_n} \int dt\, dt' \; [K_{rr'}^{(\ell)}(f, f')](t, t')
\tag{97}
$$

$$
\Theta^{(\ell+1)}(f, f') = \frac{1}{n|\operatorname{supp}(\mathcal{N}^{(\ell)}(f))|} \sum_{r, r' \in C_n} \int dt\, dt' \; [\Theta_{rr'}^{(\ell)}(f, f')](t, t').
\tag{98}
$$

*Proof.* The integral over two copies of the group in Theorem 4.3 factorize for $G = C_n \ltimes \mathbb{R}^2$ into integrals over the translations in $\mathbb{R}^2$ and sums over the discrete rotations in $C_n$. This immediately implies the recursions in the statement of the lemma. $\qquad\square$

The expressions given in the lemmata in this section can be straightforwardly implemented and therefore allow for explicit calculations of the NTK and NNGP of realistically-sized GCNNs.

# D. Equivariant NTK in the Fourier Domain for 3d Rotations

## D.1. Group Convolutions in the Fourier Domain for $G = \mathrm{SO}(3)$

For compact groups, it is possible to define a Fourier transformation. The group convolution (4) then becomes a point-wise product in the Fourier domain. For the case of $G = \mathrm{SO}(3)$, the Fourier transformation is given in terms of Wigner matrices $\mathcal{D}^l_{mn}$,

$$f(R) = \sum_{l=0}^{\infty} \frac{2l+1}{8\pi^2} \sum_{m,n=-l}^{l} \hat{f}^l_{mn} \overline{\mathcal{D}^l_{mn}(R)} \tag{99}$$

$$\hat{f}^l_{mn} = \int_{\mathrm{SO}(3)} \mathrm{d}R \ f(R) \mathcal{D}^l_{mn}(R) \,, \tag{100}$$

where $R \in \mathrm{SO}(3)$ is a rotation matrix. Note that the presented convention corresponds to the one in the `s2fft` package (Price & McEwen, 2024).

The rotations act naturally on the sphere $S^2$ on which the Fourier transform is given in terms of spherical harmonics $Y^l_m$,

$$f(x) = \sum_{l=0}^{\infty} \sum_{m=-l}^{l} \hat{f}^l_m Y^l_m(x) \tag{101}$$

$$\hat{f}^l_m = \int_{S^2} \mathrm{d}x \ f(x) \overline{Y^l_m(x)} \,. \tag{102}$$

These Fourier transformations are e.g. used in *spherical CNNs* (Cohen et al., 2018) and *steerable convolutional networks* (Weiler et al., 2018), which define equivariant group convolution layers with respect to $\mathrm{SO}(3)$ and act on input features defined on the sphere $S^2$. The change to the Fourier space is motivated by the fact that group convolutions reduce to simple multiplications of the corresponding Fourier components. For $\mathrm{SO}(3)$, the group convolutions (4) for filter support $S_\kappa \subseteq \mathrm{SO}(3)$ are defined as

$$[\mathcal{N}^{(\ell+1)}(f)](R) = \frac{1}{\sqrt{n_\ell |S_\kappa|}} \int_{S_\kappa} \mathrm{d}S \ \kappa\big(R^{-1}S\big)[\mathcal{N}^{(\ell)}(f)](S) \,. \tag{103}$$

Using

$$\mathcal{D}^l_{mn}(R^{-1}) = \overline{\mathcal{D}^l_{nm}(R)} \,, \tag{104}$$

$$\overline{\mathcal{D}^l_{mn}(R)} = (-1)^{m-n} \mathcal{D}^l_{-m,-n}(R) \,, \tag{105}$$

$$\int_{\mathrm{SO}(3)} \mathrm{d}R \ \overline{\mathcal{D}^l_{mn}(R)} \mathcal{D}^{l'}_{m'n'}(R) = \frac{8\pi^2}{2l+1} \delta_{ll'} \delta_{nn'} \delta_{mm'} \,, \tag{106}$$

the Fourier components (100) of the layer in (103) can be written compactly as

$$[\widehat{\mathcal{N}^{(\ell+1)}(f)}]^l_{mn} = \frac{1}{\sqrt{n_\ell |S_\kappa|}} \sum_{p=-l}^{l} [\widehat{\mathcal{N}^{(\ell)}(f)}]^l_{mp} \hat{\kappa}^l_{np} \,. \tag{107}$$

Note that we have assumed a real-valued kernel $\kappa$.

Similarly, the lifting layer (6) for features on $S^2$ is

$$[\mathcal{N}^{(1)}(f)](R) = \frac{1}{\sqrt{n_{\mathrm{in}} |S_\kappa|}} \int_{S^2} \mathrm{d}x \ \kappa\big(R^{-1}x\big) f(x) \,, \tag{108}$$

which, in terms of the Fourier coefficients (102) becomes

$$[\widehat{\mathcal{N}^{(1)}(f)}]^l_{mn} = \frac{1}{\sqrt{n_{\mathrm{in}} |S_\kappa|}} \frac{8\pi^2}{2l+1} \hat{f}^l_m \overline{\hat{\kappa}^l_n} \,. \tag{109}$$

Again, we have assumed a real-valued kernel $\kappa$ and used the relations

$$Y_m^l(Rx) = \sum_{n=-l}^{l} \overline{\mathcal{D}_{mn}^l(R)} Y_n^l(x), \tag{110}$$

$$\overline{Y_m^l(x)} = (-1)^m Y_{-m}^l(x), \tag{111}$$

$$\int_{S^2} \mathrm{d}x \, \overline{Y_m^l(x)} Y_{m'}^{l'}(x) = \delta_{ll'} \delta_{mm'}. \tag{112}$$

**D.2. Kernel Recursions for $G = \mathrm{SO}(3)$**

As we have seen in Section D.1 $\mathrm{SO}(3)$ group convolutions are frequently computed in the Fourier domain. In this section, we show how also the kernel recursions from Theorems 4.1 and 4.2 for group-convolution layers and lifting layers can be computed in the Fourier domain corresponding to the spherical convolutions presented in Section D.1. In the following we will assume filters $\kappa$ with global support $S_\kappa = \mathrm{SO}(3)$ or $S_\kappa = S^2$, respectively. The reason is that equations (106) and (112) otherwise have to be replaced by expressions including the Wigner's $3j$ symbols. Due to the current lack of an efficient JAX-based implementation providing their computation, we decided to restrict ourselves to the more efficient case of global filters.

In terms of the Fourier coefficients defined in (31), the recursive Kernel-relations for the spherical convolution layer (103) are specified in the following

**Lemma D.1** (Kernel recursions of $\mathrm{SO}(3)$ group-convolutions in the Fourier domain). *The layer-wise recursive relations for the NNGP and NTK of the group convolutional layer (107) for $G = \mathrm{SO}(3)$ and global filters are given by*

$$[K^{(\widehat{\ell+1})(f,f')}]_{mn,m'n'}^{l,l'} = \frac{1}{2l+1}\delta_{ll'}\delta_{n,-n'} \sum_{p=-l}^{l} (-1)^{n-p} [K^{\widehat{\ell}(f,f')}]_{mp,m'(-p)}^{l,l'} \tag{113}$$

$$[\Theta^{(\widehat{\ell+1})(f,f')}]_{mn,m'n'}^{l,l'} = [K^{(\widehat{\ell+1})(f,f')}]_{mn,m'n'}^{l,l'} + \frac{1}{2l+1}\delta_{ll'}\delta_{n,-n'} \sum_{p=-l}^{l} (-1)^{n-p} [\Theta^{\widehat{\ell}(f,f')}]_{mp,m'(-p)}^{l,l'}. \tag{114}$$

*Proof.* We identify elements of $\mathrm{SO}(3)$ with $3 \times 3$ rotation matrices $R, S, \ldots$. Then, the recursive relations for the group convolution layer from Theorem 4.1 are

$$K_{R,R'}^{(\ell+1)}(f,f') = \frac{1}{8\pi^2} \int_{\mathrm{SO}(3)} \mathrm{d}S \, K_{RS,R'S}^{(\ell)}(f,f') \tag{115}$$

$$\Theta_{R,R'}^{(\ell+1)}(f,f') = K_{R,R'}^{(\ell+1)}(f,f') + \frac{1}{8\pi^2} \int_{\mathrm{SO}(3)} \mathrm{d}S \, \Theta_{RS,R'S}^{(\ell)}(f,f'). \tag{116}$$

The Fourier coefficients of the NNGP are given by a double Fourier integral of the form (31), so the recursion (115) becomes

$$[K^{(\widehat{\ell+1})(f,f')}]_{mn,m'n'}^{l,l'} = \frac{1}{8\pi^2} \iiint_{[\mathrm{SO}(3)]^3} \mathrm{d}S \, \mathrm{d}R \, \mathrm{d}R' \, K_{RS,R'S}^{(\ell)}(f,f') \mathcal{D}_{mn}^l(R) \mathcal{D}_{m'n'}^{l'}(R'). \tag{117}$$

Plugging in the Fourier expansion of the kernel $K_{RS,R'S}^{(\ell)}(f,f')$ yields

$$[K^{(\widehat{\ell+1})(f,f')}]_{mn,m'n'}^{l,l'} = \frac{1}{8\pi^2} \iiint_{[\mathrm{SO}(3)]^3} \mathrm{d}S \, \mathrm{d}R \, \mathrm{d}R' \left( \sum_{p,p'=0}^{\infty} \frac{2p+1}{8\pi^2} \frac{2p'+1}{8\pi^2} \right.$$
$$\left. \times \sum_{q,r=-p}^{p} \sum_{q',r'=-p'}^{p'} [K^{\widehat{(\ell)}(f,f')}]_{qr,q'r'}^{p,p'} \overline{\mathcal{D}_{qr}^p(RS)\mathcal{D}_{q'r'}^{p'}(R'S)} \right) \mathcal{D}_{mn}^l(R)\mathcal{D}_{m'n'}^{l'}(R'). \tag{118}$$

Using (106), (105) and

$$\mathcal{D}_{mn}^l(RS) = \sum_{p=-l}^{l} D_{mp}^l(R)\mathcal{D}_{pn}^l(S), \tag{119}$$

we can simplify the expression to

$$[K^{\widehat{(\ell+1)}(f,f')}]^{l,l'}_{mn,m'n'} = \frac{1}{8\pi^2} \sum_{p,p'=0}^{\infty} \sum_{q,r=-p}^{p} \sum_{q',r'=-p'}^{p'} (-1)^{r-u} \frac{8\pi^2}{2l+1} \delta_{pl}\delta_{mq}\delta_{nu}\delta_{l'p'}\delta_{m'q'}\delta_{n'u'}\delta_{pp'}\delta_{u,-u'}\delta_{r,-r'}$$
$$\times [K^{\widehat{(\ell)}(f,f')}]^{p,p'}_{qr,q'r'} \tag{120}$$

$$= \frac{1}{2l+1} \delta_{ll'}\delta_{n,-n'} \sum_{r=-l}^{l} (-1)^{r-n}[K^{\widehat{(\ell)}(f,f')}]^{l,l'}_{mr,m'(-r)}. \tag{121}$$

Renaming the summation index $r \to p$ yields the desired result. The computation for the NTK is analogous. $\qquad\square$

Similarly, for the lifting layer (108) for features on the sphere, the kernel recursions can be expressed in terms of the Fourier coefficients (102) according to the following

**Lemma D.2** (Kernel recursions of spherical lifting layer in the Fourier domain)**.** *The layer-wise recursive relations for the NNGP and NTK of the lifting layer* (109) *for features on* $S^2$ *to features on* $\mathrm{SO}(3)$ *with global filters are given by*

$$[K^{\widehat{(\ell+1)}(f,f')}]^{l,l'}_{mn,m'n'} = \frac{1}{4\pi}\left(\frac{8\pi^2}{2l+1}\right)^2 (-1)^n \delta_{ll'}\delta_{n,-n'}[K^{\widehat{(\ell)}(f,f')}]^{l,l}_{m,m'} \tag{122}$$

$$[\Theta^{\widehat{(\ell+1)}(f,f')}]^{l,l'}_{mn,m'n'} = [K^{\widehat{(\ell+1)}(f,f')}]^{l,l'}_{mn,m'n'} + \frac{1}{4\pi}\left(\frac{8\pi^2}{2l+1}\right)^2 (-1)^n \delta_{ll'}\delta_{n,-n'}[\Theta^{\widehat{(\ell)}(f,f')}]^{l,l}_{m,m'}. \tag{123}$$

*Proof.* Starting from the recursive relations for the lifting layer in Theorem 4.2, the recursions in real space for $G = \mathrm{SO}(3)$ are

$$K^{(\ell+1)}_{R,R'}(f,f') = \frac{1}{4\pi}\int_{S^2} \mathrm{d}x\, K^{(\ell)}_{Rx,\,R'x}(f,f') \tag{124}$$

$$\Theta^{(\ell+1)}_{R,R'}(f,f') = K^{(\ell+1)}_{g,g'}(f,f') + \frac{1}{4\pi}\int_{S^2} \mathrm{d}x\, \Theta^{(\ell)}_{Rx,\,R'x}(f,f'). \tag{125}$$

Expressing the Fourier coefficients of the NNGP according to (31) gives

$$[K^{\widehat{(\ell+1)}(f,f')}]^{l,l'}_{mn,m'n'} = \frac{1}{4\pi}\int_{S^2} \mathrm{d}x \iint_{[\mathrm{SO}(3)]^2} \mathrm{d}R\,\mathrm{d}R'\, K^{(\ell)}_{Rx,R'x}(f,f')\mathcal{D}^l_{mn}(R)\mathcal{D}^{l'}_{m'n'}(R'). \tag{126}$$

We can now plug in the Fourier expansion of the kernel on $S^2$

$$K^{(\ell)}_{x,x'}(f,f') = \sum_{l,l'=0}^{\infty} \sum_{m=-l}^{l} \sum_{m'=-l'}^{l'} [K^{\widehat{(\ell)}(f,f')}]^{l,l'}_{m,m'}Y^l_m(x)Y^{l'}_{m'}(x'), \tag{127}$$

to obtain

$$[K^{\widehat{(\ell+1)}(f,f')}]^{l,l'}_{mn,m'n'} = \frac{1}{4\pi}\int_{S^2} \mathrm{d}x \iint_{[\mathrm{SO}(3)]^2} \mathrm{d}R\,\mathrm{d}R' \left(\sum_{p,p'=0}^{\infty} \sum_{q=-p}^{p} \sum_{q'=-p'}^{p'} [K^{\widehat{(\ell)}(f,f')}]^{p,p'}_{q,q'}Y^p_q(Rx)Y^{p'}_{q'}(R'x')\right)$$
$$\times \mathcal{D}^l_{mn}(R)\mathcal{D}^{l'}_{m'n'}(R'). \tag{128}$$

Using (106), (112), (110) and (111) one can rewrite and simplify the expression as

$$[K^{\widehat{(\ell+1)}(f,f')}]^{l,l'}_{mn,m'n'} = \frac{1}{4\pi} \sum_{p,p'=0}^{\infty} \sum_{q,r=-p}^{p} \sum_{q',r'=-p'}^{p'} \frac{8\pi^2}{2l+1}\frac{8\pi^2}{2l'+1}(-1)^{r'} \delta_{lp}\delta_{mq}\delta_{nr}\delta_{l'p'}\delta_{m'q'}\delta_{n'r'}\delta_{pp'}\delta_{r,-r'}$$
$$\times [K^{\widehat{(\ell)}(f,f')}]^{p,p'}_{q,q'} \tag{129}$$

$$= \frac{1}{4\pi}\left(\frac{8\pi^2}{2l+1}\right)^2 (-1)^n \delta_{l,l'}\delta_{n,-n'}[K^{\widehat{(\ell)}(f,f')}]^{l,l}_{m,m'}, \tag{130}$$

which is the claimed result. $\qquad\square$

# E. Proofs: Data Augmentation Versus Group Convolutions at Infinite Width

In this section, we provide proofs for the theorems given in Section 5 in the main text.

**Theorem 5.1.** *Let $\mu_t^{\text{aug}}$ and $\mu_t$ be the mean predictions after $t$ training steps of infinite ensembles of two neural network architectures $\mathcal{N}^{\text{aug}}$ and $\mathcal{N}$. Let $\mathcal{N}^{\text{aug}}$ be trained on the fully $G$-augmented training data of $\mathcal{N}$ and assume that the NTKs of the two architectures are related by*

$$\Theta(f, f') = \frac{1}{|G|} \sum_{g \in G} \Theta^{\text{aug}}(f, \rho_{\text{reg}}(g) f'). \tag{32}$$

*Then, $\mu_t^{\text{aug}}$ and $\mu_t$ converge in the infinite width limit to the same function for all $t$ for quadratic losses, up to quadratic corrections in the learning rate.*

*Proof.* For a neural network $\mathcal{N}$, we can expand the change $\Delta \mathcal{N}$ in output due to one training step of gradient descent in the learning rate $\eta$

$$\Delta \mathcal{N}_{t+1}(f) = \mathcal{N}_{t+1}(f) - \mathcal{N}_t(f) = (\theta_{t+1} - \theta_t)^\top \frac{\partial \mathcal{N}_t(f)}{\partial \theta} + \mathcal{O}(\eta^2) \tag{131}$$

$$= -\frac{\eta}{n_{\text{train}}} \sum_{i=1}^{n_{\text{train}}} \underbrace{\left(\frac{\partial \mathcal{N}_t(f)}{\partial \theta}\right)^\top \frac{\partial \mathcal{N}_t(f_i)}{\partial \theta}}_{\Theta_t(f, f_i)} \mathcal{L}'(\mathcal{N}_t(f_i), y_i) + \mathcal{O}(\eta^2), \tag{132}$$

where $\Theta_t$ is the empirical NTK at training step $t$, $y_i$ are the training labels and $\mathcal{L}'$ is the derivative of the per-sample loss with respect to the output of the network. Taking the mean and the infinite width limit yields

$$\Delta \mu_{t+1}(f) = -\frac{\eta}{n_{\text{train}}} \sum_{i=1}^{n_{\text{train}}} \Theta(f, f_i) \mathcal{L}'(\mu_t(f_i), y_i), \tag{133}$$

since we have assumed that $\mathcal{L}'$ is linear in its first argument.

The network $\mathcal{N}^{\text{aug}}$ on the other hand is trained using full data augmentation over $G$, so we can decompose the sum over training samples into a sum over the training samples in (133) and a sum over $G$. Note that since we assume full data augmentation and a finite training set, we restrict to $G$ being finite in this section. We obtain

$$\Delta \mu_{t+1}^{\text{aug}}(f) = -\frac{\eta}{n_{\text{train}}|G|} \sum_{g \in G} \sum_{i=1}^{n_{\text{train}}} \Theta^{\text{aug}}(f, \rho_{\text{reg}}(g) f_i) \mathcal{L}'(\mu_t^{\text{aug}}(\rho_{\text{reg}}(g) f_i), y_i). \tag{134}$$

As mentioned in the main text, we will prove the statement inductively over training steps $t$. At $t = 0$, the mean output of all neural networks is zero in the infinite width limit (Neal, 1996; Lee et al., 2018). For the induction step, assume that $\mu_t^{\text{aug}} = \mu_t$. Then, $\mu_{t+1}^{\text{aug}} = \mu_{t+1}$ if $\Delta \mu_{t+1}^{\text{aug}} = \Delta \mu_{t+1}$. Since the ensemble mean of networks trained with data augmentation is exactly equivariant (Gerken & Kessel, 2024; Nordenfors & Flinth, 2024), we have $\mu_t^{\text{aug}}(\rho_{\text{reg}}(g) f_i) = \mu_t^{\text{aug}}(f_i) = \mu_t(f_i)$ by the induction assumption. Therefore,

$$\Delta \mu_{t+1}^{\text{aug}}(f) = -\frac{\eta}{n_{\text{train}}|G|} \sum_{g \in G} \sum_{i=1}^{n_{\text{train}}} \Theta^{\text{aug}}(f, \rho_{\text{reg}}(g) f_i) \mathcal{L}'(\mu_t(f_i), y_i). \tag{135}$$

Using assumption (32) concludes the proof,

$$\Delta \mu_{t+1}^{\text{aug}}(f) = -\eta \sum_{i=1}^{n_{\text{train}}} \Theta(f, f_i) \mathcal{L}'(\mu_t(f_i), y_i) = \Delta \mu_{t+1}(f). \tag{136}$$

$\square$

As mentioned in Section 7, Theorem 5.1 can be generalized to equivariantly data-augmented networks trained on

$$\bigcup_{i=1}^{n_{\text{train}}} \bigcup_{g \in G} \{(\rho_{\text{reg}}(g)f_i, \tilde{\rho}_{\text{reg}}(g)y_i)\}, \tag{137}$$

where the targets $y_i : G \to \mathbb{R}^{n_{\text{out}}}$ are signals on the group.

**Theorem E.1.** *Let $\mu_t^{\text{aug}}$ and $\mu_t$ be the mean predictions after $t$ training steps of infinite ensembles of two neural network architectures $\mathcal{N}^{\text{aug}}$ and $\mathcal{N}$. Let $\mathcal{N}^{\text{aug}}$ be trained on the fully equivariantly $G$-augmented training data of $\mathcal{N}$ and assume that the NTKs of the two architectures are related by*

$$\Theta_{g,g'}(f, f') = \frac{1}{|G|} \sum_{h \in g} \Theta_{g,hg'}^{\text{aug}}(f, \rho_{\text{reg}}(h)f'). \tag{138}$$

*Then, $\mu_t^{\text{aug}}$ and $\mu_t$ converge in the infinite width limit to the same function for all $t$ for quadratic losses, up to quadratic corrections in the learning rate.*

*Proof.* This theorem is a straightforward extension of Theorem 5.1, which is why we only highlight the differences. Similarly to (132), the change in the output after one training step on unaugmented data is given by

$$[\Delta\mathcal{N}_{t+1}(f)](g) = -\frac{\eta}{n_{\text{train}}} \sum_{i=1}^{n_{\text{train}}} \sum_{g' \in G} \Theta_{t;g,g'}(f, f_i)\mathcal{L}'([\mathcal{N}_t(f_i)](g'), y_i(g')) + \mathcal{O}(\eta^2), \tag{139}$$

where $\mathcal{L}$ is the pointwise per-sample loss. Again, $\mathcal{L}'$ is the derivative with respect to the output of the network at a given point. As before, we assume that $\mathcal{L}'$ is linear in its first argument, allowing us to simplify the expectation of the infinite-width version of (139) to

$$[\Delta\mu_{t+1}(f)](g) = -\frac{\eta}{n_{\text{train}}} \sum_{i=1}^{n_{\text{train}}} \sum_{g' \in G} \Theta_{g,g'}(f, f_i)\,\mathcal{L}'([\mu_t(f_i)](g'), y_i(g')). \tag{140}$$

In a similar fashion, we derive the update of a network trained on fully equivariantly augmented data and obtain

$$[\Delta\mu_{t+1}^{\text{aug}}(f)](g) = -\frac{\eta}{n_{\text{train}}|G|} \sum_{g',h \in G} \sum_{i=1}^{n_{\text{train}}} \Theta_{g,g'}^{\text{aug}}(f, \rho_{\text{reg}}(h)f_i)\mathcal{L}'([\mu_t^{\text{aug}}(\rho_{\text{reg}}(h)f_i)](g'), [\tilde{\rho}_{\text{reg}}(h)y_i](g')). \tag{141}$$

Using the equivariance property of the ensemble mean again (Gerken & Kessel, 2024), i.e. $\mu_t^{\text{aug}}(\rho_{\text{reg}}(h)f_i) = \tilde{\rho}_{\text{reg}}(h)\mu_t^{\text{aug}}(f_i)$, and shifting the summation as $g' \to hg'$, we obtain

$$[\Delta\mu_{t+1}^{\text{aug}}(f)](g) = -\frac{\eta}{n_{\text{train}}|G|} \sum_{g',h \in G} \sum_{i=1}^{n_{\text{train}}} \Theta_{g,hg'}^{\text{aug}}(f, \rho_{\text{reg}}(h)f_i)\mathcal{L}'([\mu_t^{\text{aug}}(f_i)](g'), y_i(g')) \tag{142}$$

$$= -\frac{\eta}{n_{\text{train}}} \sum_{g' \in G} \sum_{i=1}^{n_{\text{train}}} \left( \frac{1}{|G|} \sum_{h \in G} \Theta_{g,hg'}^{\text{aug}}(f, \rho_{\text{reg}}(h)f_i) \right) \mathcal{L}'([\mu_t^{\text{aug}}(f_i)](g'), y_i(g')), \tag{143}$$

where we have used that $[\mu_t^{\text{aug}}(\rho_{\text{reg}}(h)f_i)](hg') = [\mu_t^{\text{aug}}(f_i)](g')$ and analogously for $y_i$.

Using (138) and following the same inductive argument as in the proof of Theorem 5.1 concludes this proof. $\square$

**Theorem 5.2.** *Let $\mathcal{N}^{\text{FC}}$ be an MLP acting on feature maps with output in $\mathbb{R}$ and architecture*

$$\mathcal{N}^{\text{FC}} = \text{FC}^{(L)} \circ \sigma \circ \cdots \circ \text{FC}^{(3)} \circ \sigma \circ \text{FC}^{(1)}, \tag{33}$$

*where FC denotes a dense MLP layer and $\sigma$ a point-wise nonlinearity. Let $\mathcal{N}^{\text{GC}}$ be a $G$-invariant GCNN with architecture*

$$\mathcal{N}^{\text{GC}} = \text{GPool} \circ \text{GConv}(S_\kappa^L) \circ \sigma \circ \text{GConv}(S_\kappa^{L-2}) \circ \sigma \cdots$$
$$\cdots \circ \text{GConv}(S_\kappa^3) \circ \sigma \circ \text{Lifting}(S_\kappa^1), \tag{34}$$

*where $S^\ell_\kappa$ are the supports of the convolutional filters with $S^1_\kappa = X$, the domain of the input feature maps, and the other $S^\ell_\kappa$ are invariant under $G$. Then, the $G$-averages of the kernels of the MLP are given by the kernels of the GCNN,*

$$K^{\mathrm{GC}}(f, f') = \frac{1}{\mathrm{vol}(G)} \int \mathrm{d}g \; K^{\mathrm{FC}}(f, \rho_{\mathrm{reg}}(g)f') \tag{35}$$

$$\Theta^{\mathrm{GC}}(f, f') = \frac{1}{\mathrm{vol}(G)} \int \mathrm{d}g \; \Theta^{\mathrm{FC}}(f, \rho_{\mathrm{reg}}(g)f') \,. \tag{36}$$

*Proof.* In order to proof the kernel equalities, we will construct the kernels for the fully connected architecture (33) and the group convolutional architecture (34) by explicitly iterating the recursion relations.

The iteration starts with the input kernels which for the fully-connected network are

$$K^{\mathrm{FC}(0)}(f, f') = \frac{1}{\mathrm{vol}(X)} \int \mathrm{d}x \; f(x)f'(x) \,, \qquad \Theta^{\mathrm{FC}(0)}(f, f') = 0 \,, \tag{144}$$

since the different points in the domain $X$ of the input function take the role of different channels when the image tensor is flattened. The first layer of the FC-network is a fully-connected layer. These update the kernels according to (Jacot et al., 2018)

$$K^{\mathrm{FC}(\ell+1)}(f, f') = K^{\mathrm{FC}(\ell)}(f, f') \tag{145}$$

$$\Theta^{\mathrm{FC}(\ell+1)}(f, f') = K^{\mathrm{FC}(\ell+1)}(f, f') + \Theta^{\mathrm{FC}(\ell)}(f, f') \,. \tag{146}$$

In order to write kernel transformation like this more compactly, we will collect all relevant kernels at layer $\ell$ into an $\mathbb{R}^4$ vector $\Xi^{\mathrm{FC}(\ell)}(f, f')$ according to

$$\Xi^{\mathrm{FC}(\ell)}(f, f') = \begin{pmatrix} K^{\mathrm{FC}(\ell)}(f, f) \\ K^{\mathrm{FC}(\ell)}(f, f') \\ K^{\mathrm{FC}(\ell)}(f', f') \\ \Theta^{\mathrm{FC}(\ell)}(f, f') \end{pmatrix} \,, \tag{147}$$

where the components $K^{\mathrm{FC}(\ell)}(f, f)$ and $K^{\mathrm{FC}(\ell)}(f', f')$ are needed for the nonlinear layers below. In the $\Xi^{\mathrm{FC}}$-notation, (145), (146) can be summarized by a function $\mathcal{G} : \mathbb{R}^4 \to \mathbb{R}^4$ mapping $\Xi^{\mathrm{FC}(\ell)}(f, f') \mapsto \Xi^{\mathrm{FC}(\ell+1)}(f, f')$, defined by

$$\mathcal{G} \begin{pmatrix} k_1 \\ k_2 \\ k_3 \\ \Theta \end{pmatrix} = \begin{pmatrix} k_1 \\ k_2 \\ k_3 \\ k_2 + \Theta \end{pmatrix} \,. \tag{148}$$

Therefore, the kernels of the first fully-connected layer take the form

$$\Xi^{\mathrm{FC}(1)}(f, f') = \mathcal{G}(\Xi^{\mathrm{FC}(0)}(f, f')) = \frac{1}{\mathrm{vol}(X)} \int \mathrm{d}x \begin{pmatrix} f(x)f(x) \\ f(x)f'(x) \\ f'(x)f'(x) \\ f(x)f'(x) \end{pmatrix} \,. \tag{149}$$

In the architecture (33), fully connected layers are alternated with nonlinearities, which act according to (Jacot et al., 2018)

$$\Lambda^{\mathrm{FC}(\ell)}(f, f') = \begin{pmatrix} K^{\mathrm{FC}(\ell)}(f, f) & K^{\mathrm{FC}(\ell)}(f, f') \\ K^{\mathrm{FC}(\ell)}(f', f) & K^{\mathrm{FC}(\ell)}(f', f') \end{pmatrix} \tag{150}$$

$$K^{\mathrm{FC}(\ell+1)}(f, f') = \mathbb{E}_{(u,v)\sim\mathcal{N}(0, \Lambda^{\mathrm{FC}(\ell)}(f,f'))}[\sigma(u)\sigma(v)] \tag{151}$$

$$\dot{K}^{\mathrm{FC}(\ell+1)}(f, f') = \mathbb{E}_{(u,v)\sim\mathcal{N}(0, \Lambda^{\mathrm{FC}(\ell)}(f,f'))}[\sigma'(u)\sigma'(v)] \tag{152}$$

$$\Theta^{\mathrm{FC}(\ell+1)}(f, f') = \dot{K}^{\mathrm{FC}(\ell+1)}(f, f')\Theta^{\mathrm{FC}(\ell)}(f, f') \,. \tag{153}$$

on the kernels. We will denote the corresponding action on the $\Xi^{\mathrm{FC}}$-vectors by a function $\mathcal{F}_\sigma : \mathbb{R}^4 \to \mathbb{R}^4$. Therefore, a fully-connected layer followed by a nonlinearity can be written as

$$\Xi^{\mathrm{FC}(\ell+2)}(f, f') = \mathcal{F}_\sigma(\mathcal{G}(\Xi^{\mathrm{FC}(\ell)}(f, f'))) \,. \tag{154}$$

Hence, in this notation, the kernels of the entire FC network are given by

$$\Xi^{\text{FC}}(f, f') = \mathcal{G}\Big(\mathcal{F}_\sigma\Big(\cdots \mathcal{G}\Big(\mathcal{F}_\sigma\Big(\mathcal{G}(\Xi^{\text{FC}(0)}(f, f'))\Big)\Big)\cdots\Big)\Big). \tag{155}$$

Next, we compute the kernels of the GCNN. The input kernels in this case are

$$K^{\text{GC}(0)}_{x,x'}(f, f') = f(x)f'(x), \qquad \Theta^{\text{GC}(0)}_{x,x'}(f, f') = 0. \tag{156}$$

According to (34), the first layer of the network is a lifting layer whose recursion relation was given in Theorem 4.2. Again, we define an $\mathbb{R}^4$-vector to collect all kernel components necessary for computing the kernels of the network,

$$\Xi^{\text{GC}(\ell)}_{g,g'}(f, f') = \begin{pmatrix} K^{\text{GC}(\ell)}_{g,g}(f, f) \\ K^{\text{GC}(\ell)}_{g,g'}(f, f') \\ K^{\text{GC}(\ell)}_{g',g'}(f', f') \\ \Theta^{\text{GC}(\ell)}_{g,g'}(f, f') \end{pmatrix}. \tag{157}$$

In terms of $\Xi^{\text{GC}}$ the kernels of the lifting layer are given by (note that the filter of the lifting layer has global support by assumption)

$$\Xi^{\text{GC}(1)}_{g,g'}(f, f') = \mathcal{G}\left(\frac{1}{\text{vol}(X)}\int_X \mathrm{d}x \begin{pmatrix} K^{\text{GC}(0)}_{\rho(g)x,\rho(g)x}(f, f) \\ K^{\text{GC}(0)}_{\rho(g)x,\rho(g')x}(f, f') \\ K^{\text{GC}(0)}_{\rho(g')x,\rho(g')x}(f', f') \\ \Theta^{\text{GC}(0)}_{\rho(g)x,\rho(g')x}(f, f') \end{pmatrix}\right) = \frac{1}{\text{vol}(X)}\int_X \mathrm{d}x \begin{pmatrix} f(\rho(g)x)f(\rho(g)x) \\ f(\rho(g)x)f'(\rho(g')x) \\ f'(\rho(g')x)f'(\rho(g')x) \\ f(\rho(g)x)f'(\rho(g')x) \end{pmatrix}. \tag{158}$$

For later convenience, we note here that

$$\Xi^{\text{GC}(1)}_{h,g^{-1}h}(f, f') = \frac{1}{\text{vol}(X)}\int_X \mathrm{d}x \begin{pmatrix} f(\rho(h)x)f(\rho(h)x) \\ f(\rho(h)x)f'(\rho(g^{-1}h)x) \\ f'(\rho(g^{-1}h)x)f'(\rho(g^{-1}h)x) \\ f(\rho(h)x)f'(\rho(g^{-1}h)x) \end{pmatrix} \tag{159}$$

$$= \frac{1}{\text{vol}(X)}\int_X \mathrm{d}x \begin{pmatrix} f(x)f(x) \\ f(x)f'(\rho(g^{-1})x) \\ f'(\rho(g^{-1})x)f'(\rho(g^{-1})x) \\ f(x)f'(\rho(g^{-1})x) \end{pmatrix} \tag{160}$$

$$= \Xi^{\text{FC}(1)}(f, \rho_{\text{reg}}(g)f'), \tag{161}$$

where we shifted the integration variable in the second step and used (149).

After the lifting layer, we act with a point-wise nonlinearty whose recursion relations are given in Corollary 4.4. Since this transformation is independent for the different $g, g'$-components, we can write it using the same function $\mathcal{F}_\sigma$ introduced above as

$$\Xi^{\text{GC}(\ell+1)}_{g,g'}(f, f') = \mathcal{F}_\sigma(\Xi^{\text{GC}(\ell)}_{g,g'}(f, f')). \tag{162}$$

A GCNN layer transforms the NNGP and NTK according to Theorem 4.1. We can write this in terms of $\Xi^{\text{GC}(\ell)}_{g,g'}$ as

$$\Xi^{\text{GC}(\ell+1)}_{g,g'}(f, f') = \frac{1}{|S^\ell_\kappa|}\int_{S_\kappa} \mathrm{d}h_\ell \, \mathcal{G}(\Xi^{\text{GC}(\ell)}_{gh_\ell,g'h_\ell}(f, f')), \tag{163}$$

with $\mathcal{G}$ as introduced in (148). The final pooling layer acts according to Theorem 4.3, which we can write as

$$\Xi^{\text{GC}(\ell)}_{g,g'}(f, f') = \frac{1}{(\text{vol}(G))^2}\int_G \mathrm{d}g \int_G \mathrm{d}g' \, \Xi^{\text{GC}(\ell)}_{g,g'}(f, f'). \tag{164}$$

With the expressions (162), (163) and (164), we can write the kernels of the entire network as

$$\Xi^{\text{GC}}(f, f') = \frac{1}{(\text{vol}(G))^2} \int_G \mathrm{d}g \int_G \mathrm{d}g' \, \frac{1}{|S_\kappa^L|} \int_{S_\kappa^L} \mathrm{d}h_L \, \mathcal{G}\left(\mathcal{F}_\sigma\left(\frac{1}{|S_\kappa^{L-2}|}\int_{S_\kappa^{L-2}} \mathrm{d}h_{L-2} \, \mathcal{G}\left(\mathcal{F}_\sigma\left(\cdots\right.\right.\right.\right.$$
$$\left.\left.\left.\left.\cdots \frac{1}{|S_\kappa^3|}\int_{S_\kappa^3} \mathrm{d}h_3 \, \mathcal{G}\left(\mathcal{F}_\sigma(\Xi^{\text{GC}(1)}_{gh_L h_{L-2}\cdots h_5 h_3, g' h_L h_{L-2}\cdots h_5 h_3}(f,f'))\right)\cdots\right)\right)\right)\right). \quad (165)$$

In order to simplify this expression, we shift $h_3$ and absorb $gh_L h_{L-2}\cdots h_5$ into it. This will not change the integration domain of $h_3$ since $S_\kappa^3$ is by assumption invariant under $G$. Then, the integrals over $h_L, h_{L-2}, \ldots, h_5$ become trivial and cancel against their $1/|S_\kappa^\ell|$-prefactors. We are left with

$$\Xi^{\text{GC}}(f, f') = \frac{1}{(\text{vol}(G))^2} \int_G \mathrm{d}g \int_G \mathrm{d}g' \, \mathcal{G}\left(\mathcal{F}_\sigma\left(\mathcal{G}\left(\mathcal{F}_\sigma\left(\cdots \frac{1}{|S_\kappa^3|}\int_{S_\kappa^3} \mathrm{d}h_3 \, \mathcal{G}\left(\mathcal{F}_\sigma(\Xi^{\text{GC}(1)}_{h_3, g'g^{-1}h_3}(f,f'))\right)\cdots\right)\right)\right)\right). \quad (166)$$

Finally, we trivialize the $g'$-integral by shifting $g^{-1}$ to absorb $g'$. Thus, we obtain

$$\Xi^{\text{GC}}(f, f') = \frac{1}{\text{vol}(G)} \int_G \mathrm{d}g \, \mathcal{G}\left(\mathcal{F}_\sigma\left(\mathcal{G}\left(\mathcal{F}_\sigma\left(\cdots \frac{1}{|S_\kappa^3|}\int_{S_\kappa^3} \mathrm{d}h_3 \, \mathcal{G}\left(\mathcal{F}_\sigma(\Xi^{\text{GC}(1)}_{h_3, g^{-1}h_3}(f,f'))\right)\cdots\right)\right)\right)\right) \quad (167)$$

$$= \frac{1}{\text{vol}(G)} \int_G \mathrm{d}g \, \mathcal{G}\left(\mathcal{F}_\sigma\left(\mathcal{G}\left(\mathcal{F}_\sigma\left(\cdots \mathcal{G}\left(\mathcal{F}_\sigma(\Xi^{\text{FC}(1)}(f, \rho_{\text{reg}}(g)f'))\right)\cdots\right)\right)\right)\right) \quad (168)$$

$$= \frac{1}{\text{vol}(G)} \int_G \mathrm{d}g \, \Xi^{\text{FC}}(f, \rho_{\text{reg}}(g)f'), \quad (169)$$

where we used (161), trivializing the integral over $h_3$, and then identified $\Xi^{\text{FC}}$ from (155). The statement follows by taking the second and fourth components of (169). $\qquad\square$

**Theorem 5.3.** *Let $\mathcal{N}^{K\ltimes N}$ be the $K \ltimes N$-invariant GCNN with architecture (34) and $K$-invariant filter supports $S_\kappa^\ell$ which for the GConv-layers decompose as $S_\kappa^\ell = K_\kappa^\ell \times N_\kappa^\ell$, $K_\kappa^\ell \subseteq K$, $N_\kappa^\ell \subseteq N$. Let $\mathcal{N}^N$ be the $N$-invariant GCNN with architecture (34) and filter supports $N_\kappa^L, \ldots, N_\kappa^3$ and $S_\kappa^1$. Then, the NNGPs and NTKs of these networks are related by*

$$K^{K\ltimes N}(f, f') = \frac{1}{\text{vol}(K)} \int_K \mathrm{d}k \, K^N(f, \rho_{\text{reg}}(k)f') \quad (37)$$

$$\Theta^{K\ltimes N}(f, f') = \frac{1}{\text{vol}(K)} \int_K \mathrm{d}k \, \Theta^N(f, \rho_{\text{reg}}(k)f'). \quad (38)$$

*Proof.* In this proof, we will use the same notation as in the proof for Theorem 5.2 above and use results from there as well. We start by considering the kernels $\Xi^{K\ltimes N}$ of $\mathcal{N}^{K\ltimes N}$ by specializing (165) to the case $G = K \ltimes N$. Due to the semidirect product structure of $G$, there is a unique decomposition $g = kn$ for each $g \in G$ into $k \in K$ and $n \in N$. Since by assumption the filter supports $S_\kappa^\ell$ on $G$ also factorize over $K$ and $N$, we can split all $G$-integrations in (165) over $N$ and $K$ and obtain

$$\Xi^{K\ltimes N}(f, f') = \frac{1}{(\text{vol}(K))^2}\frac{1}{(\text{vol}(N))^2} \int_K \mathrm{d}k \int_N \mathrm{d}n \int_K \mathrm{d}k' \int_N \mathrm{d}n' \, \frac{1}{|K_\kappa^L||N_\kappa^L|} \int_{K_\kappa^L} \mathrm{d}j_L \int_{N_\kappa^L} \mathrm{d}m_L \, \mathcal{G}\left(\mathcal{F}_\sigma \cdots\right.$$
$$\left.\cdots \frac{1}{|K_\kappa^3||N_\kappa^3|} \int_{K_\kappa^3} \mathrm{d}j_3 \int_{N_\kappa^3} \mathrm{d}m_3 \, \mathcal{G}\left(\mathcal{F}_\sigma(\Xi^{K\ltimes N(1)}_{knj_L m_L\cdots j_3 m_3, k'n'j_L m_L\cdots j_3 m_3}(f,f'))\right)\cdots\right). \quad (170)$$

In order to trivialize the integrals over $K$, as was done with the integrals over $G$ in (166), we need to rewrite the first group

index of $\Xi^{K \ltimes N(1)}$ such that all $j_\ell$ appear next to each other. To this end, we introduce several unit elements

$$knj_Lm_L \cdots j_7m_7j_5m_5j_3m_3 = knj_Lm_L \cdots j_7m_7j_5j_3 \underbrace{j_3^{-1}m_5j_3}_{\in N} m_3 \tag{171}$$

$$= knj_Lm_L \cdots j_7j_5j_3 \underbrace{(j_5j_3)^{-1}m_7j_5j_3}_{\in N} \underbrace{j_3^{-1}m_5j_3}_{\in N} m_3 \tag{172}$$

$$\vdots$$

$$= kj_L \cdots j_3 \underbrace{(j_L \cdots j_3)^{-1}nj_L \cdots j_3}_{\in N} \underbrace{(j_{L-2} \cdots j_3)^{-1}m_L \cdots}_{\in N} \underbrace{j_3^{-1}m_5j_3}_{\in N} m_3 \,. \tag{173}$$

We perform the same rewriting also on the second group index of $\Xi^{K \ltimes N(1)}$. Since $N$ is a normal subgroup of $G$, $knk^{-1} \in N$ for all $k \in K$, $n \in N$ and the Haar measure on $N$ is invariant under shifts of the form $n \to knk^{-1}$. Furthermore, the integration domains $N_\kappa^\ell$ are by assumption invariant under this transformation. Hence, we shift $n$, $n'$ and the $m_\ell$ by

$$n \to j_L \cdots j_3n(j_L \cdots j_3)^{-1} \tag{174}$$

$$n' \to j_L \cdots j_3n'(j_L \cdots j_3)^{-1} \tag{175}$$

$$m_\ell \to j_{\ell-2} \cdots j_3m_\ell(j_{\ell-2} \cdots j_3)^{-1} \qquad \ell > 3 \,. \tag{176}$$

With this (170) becomes

$$\Xi^{K \ltimes N}(f, f') = \frac{1}{(\text{vol}(K))^2} \frac{1}{(\text{vol}(N))^2} \int_K \mathrm{d}k \int_N \mathrm{d}n \int_K \mathrm{d}k' \int_N \mathrm{d}n' \frac{1}{|K_\kappa^L||N_\kappa^L|} \int_{K_\kappa^L} \mathrm{d}j_L \int_{N_\kappa^L} \mathrm{d}m_L \, \mathcal{G}\bigg(\mathcal{F}_\sigma \cdots$$

$$\cdots \frac{1}{|K_\kappa^3||N_\kappa^3|} \int_{K_\kappa^3} \mathrm{d}j_3 \int_{N_\kappa^3} \mathrm{d}m_3 \, \mathcal{G}\left(\mathcal{F}_\sigma(\Xi^{K \ltimes N(1)}_{kj_L \cdots j_3nm_L \cdots m_3,k'j_L \cdots j_3n'm_L \cdots m_3}(f, f'))\right) \cdots \bigg) \tag{177}$$

$$= \frac{1}{(\text{vol}(K))^2} \frac{1}{(\text{vol}(N))^2} \int_K \mathrm{d}k \int_N \mathrm{d}n \int_K \mathrm{d}k' \int_N \mathrm{d}n' \frac{1}{|N_\kappa^L|} \int_{N_\kappa^L} \mathrm{d}m_L \, \mathcal{G}\bigg(\mathcal{F}_\sigma \cdots$$

$$\cdots \frac{1}{|K_\kappa^3||N_\kappa^3|} \int_{K_\kappa^3} \mathrm{d}j_3 \int_{N_\kappa^3} \mathrm{d}m_3 \, \mathcal{G}\left(\mathcal{F}_\sigma(\Xi^{K \ltimes N(1)}_{j_3nm_L \cdots m_3,k'k^{-1}j_3n'm_L \cdots m_3}(f, f'))\right) \cdots \bigg) \tag{178}$$

$$= \frac{1}{\text{vol}(K)} \frac{1}{(\text{vol}(N))^2} \int_K \mathrm{d}k \int_N \mathrm{d}n \int_N \mathrm{d}n' \frac{1}{|N_\kappa^L|} \int_{N_\kappa^L} \mathrm{d}m_L \, \mathcal{G}\bigg(\mathcal{F}_\sigma \cdots$$

$$\cdots \frac{1}{|K_\kappa^3||N_\kappa^3|} \int_{K_\kappa^3} \mathrm{d}j_3 \int_{N_\kappa^3} \mathrm{d}m_3 \, \mathcal{G}\left(\mathcal{F}_\sigma(\Xi^{K \ltimes N(1)}_{j_3nm_L \cdots m_3,k^{-1}j_3n'm_L \cdots m_3}(f, f'))\right) \cdots \bigg) \,. \tag{179}$$

Here, we shifted $j_3 \to (kj_L \cdots j_5)$ in the first step, trivializing the integrals over $j_L, \ldots, j_5$ which then cancel against their $1/|K_\kappa^\ell|$-prefactors. In the second step, we first trivialized the integral over $k'$ by shifting $k' \to k'k$ and then canceled it against its $1/\text{vol}(K)$-prefactor.

Next, we perform another manipulation on the group indices of $\Xi^{K \ltimes N(1)}$ by first inserting suitable unit elements,

$$j_3nm_L \cdots m_3 = \underbrace{j_3nj_3^{-1}}_{\in N} \underbrace{j_3m_Lj_3^{-1}}_{\in N} \underbrace{j_3m_{L-2}j_3^{-1}}_{\in N} \cdots \underbrace{j_3m_3j_3^{-1}}_{\in N} j_3 \,, \tag{180}$$

and similarly for the second group index of $\Xi^{K \ltimes N(1)}$. After shifting

$$n \to j_3^{-1}nj_3 \,, \qquad n' \to j_3^{-1}n'j_3 \,, \qquad m_\ell \to j_3^{-1}m_\ell j_3 \quad \ell \geq 3 \,, \tag{181}$$

in (179), we obtain

$$\Xi^{K \ltimes N}(f, f') = \frac{1}{\text{vol}(K)} \frac{1}{(\text{vol}(N))^2} \int_K \mathrm{d}k \int_N \mathrm{d}n \int_N \mathrm{d}n' \frac{1}{|N_\kappa^L|} \int_{N_\kappa^L} \mathrm{d}m_L \, \mathcal{G}\bigg(\mathcal{F}_\sigma \cdots$$

$$\cdots \frac{1}{|K_\kappa^3||N_\kappa^3|} \int_{K_\kappa^3} \mathrm{d}j_3 \int_{N_\kappa^3} \mathrm{d}m_3 \, \mathcal{G}\left(\mathcal{F}_\sigma(\Xi^{K \ltimes N(1)}_{nm_L \cdots m_3j_3,k^{-1}n'm_L \cdots m_3j_3}(f, f'))\right) \cdots \bigg) \,. \tag{182}$$

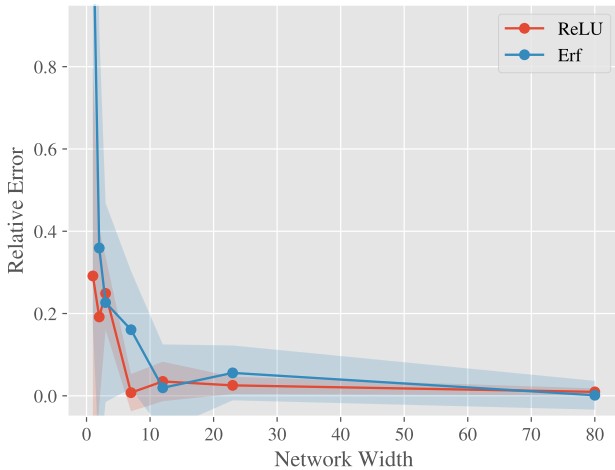

Figure 5. **Convergence of the Monte-Carlo estimates of the NNGP to their infinite-width limits for** $G = C_4 \ltimes \mathbb{R}^2$**.** Plotted is the relative error averaged over the components of a $3 \times 3$ Gram matrix for networks with a ReLU or an error function nonlinearity. The bands correspond to $\pm$ one standard deviation of the estimator.

As in the proof for Theorem 5.2, we will now write $\Xi^{K \ltimes N(1)}$ in terms of $\Xi^{N(1)}$. Using the shorthand $\tilde{m} = m_L \cdots m_3$ and the analogous steps to (161), we find

$$\Xi^{K \ltimes N(1)}_{n\tilde{m}j_3, k^{-1}n'\tilde{m}j_3}(f, f') = \frac{1}{|S^1_\kappa|} \int_{S^1_\kappa} \mathrm{d}x \begin{pmatrix} f(\rho(n\tilde{m}j_3)x)f(\rho(n\tilde{m}j_3)x) \\ f(\rho(n\tilde{m}j_3)x)f'(\rho(k^{-1}n'\tilde{m}j_3)x) \\ f'(\rho(k^{-1}n'\tilde{m}j_3)x)f'(\rho(k^{-1}n'\tilde{m}j_3)x) \\ f(\rho(n\tilde{m}j_3)x)f'(\rho(k^{-1}n'\tilde{m}j_3)x) \end{pmatrix} \tag{183}$$

$$= \frac{1}{|S^1_\kappa|} \int_{S^1_\kappa} \mathrm{d}x \begin{pmatrix} f(\rho(n\tilde{m})x)f(\rho(n\tilde{m})x) \\ f(\rho(n\tilde{m})x)f'(\rho(k^{-1}n'\tilde{m})x) \\ f'(\rho(k^{-1}n'\tilde{m})x)f'(\rho(k^{-1}n'\tilde{m})x) \\ f(\rho(n\tilde{m})x)f'(\rho(k^{-1}n'\tilde{m})x) \end{pmatrix} \tag{184}$$

$$= \Xi^{N(1)}_{n\tilde{m}, n'\tilde{m}}(f, \rho_{\mathrm{reg}}(k)f') , \tag{185}$$

where for the second equality, we have shifted $x \to \rho(j_3^{-1})x$, which leaves $S^1_\kappa$ invariant by assumption. Plugging (185) into (182) trivializes the $j_3$-integral which cancels against its $1/|K^3_\kappa|$-prefactor, yielding

$$\Xi^{K \ltimes N}(f, f') = \frac{1}{\mathrm{vol}(K)} \int_K \mathrm{d}k \; \frac{1}{(\mathrm{vol}(N))^2} \int_N \mathrm{d}n \int_N \mathrm{d}n' \; \frac{1}{|N^L_\kappa|} \int_{N^L_\kappa} \mathrm{d}m_L \; \mathcal{G}\left(\mathcal{F}_\sigma \cdots \right.$$

$$\left. \cdots \frac{1}{|N^3_\kappa|} \int_{N^3_\kappa} \mathrm{d}m_3 \; \mathcal{G}\left(\mathcal{F}_\sigma(\Xi^{N(1)}_{nm_L\cdots m_3, n'm_L\cdots m_3}(f, \rho_{\mathrm{reg}}(k)f'))\right)\cdots\right) \tag{186}$$

$$= \frac{1}{\mathrm{vol}(K)} \int_K \mathrm{d}k \; \Xi^N(f, \rho_{\mathrm{reg}}(k)f') , \tag{187}$$

where we have identified $\Xi^N$ by comparing to (165). The statement of the theorem follows by considering the second and fourth components of (187). $\qquad\square$

# F. Further Experimental Results

In this appendix, we provide further details and results of the numerical experiments presented in Section 6.

## F.1. Kernel Convergence

Figure 5 shows the convergence of Monte-Carlo estimates of the NNGP to the analytical infinite-width expression derived using the theorems in Section 4.1.

*Table 1.* Architectures used for the medical image classification described in Section 6. For convolutional, group-convolutional and lifting layers, the argument is the kernel size (all kernels are squared). Both pooling layers are global. The number of output neurons is finite and has to correspond to the 9 classes.

| CNN | GCNN |
| --- | --- |
| Conv(3) | Lifting(3) |
| ReLU | ReLU |
| Conv(3) | GConv(3) |
| ReLU | ReLU |
| Conv(3) | GConv(3) |
| ReLU | ReLU |
| Conv(3) | GConv(3) |
| ReLU | ReLU |
| Conv(3) | GConv(3) |
| ReLU | ReLU |
| SumPool | GPool |
| Dense | Dense |
| ReLU | ReLU |
| Dense(9) | Dense(9) |

## F.2. Medical Image Experiments

In the infinite-width limit, the NTK becomes deterministic and time-independent under the gradient flow dynamics. In the case of MSE loss, the differential equation describing the mean output of a network at time $t$ becomes a linear ODE, thus allowing for an analytic expression at arbitrary time. In the limit of infinite training time, the mean is given by (41) (Jacot et al., 2018). This relation is effectively a kernel method that can be used to generate prediction of the infinitely wide network.

The task consists of classifying histological images (Kather et al., 2018) containing nine classes of tissues, two of which are cancerous. The original images have a resolution of $224 \times 224$ pixels each and have been down-scaled to a resolution of $32 \times 32$ pixels to reduce the kernel evaluation time. Note that the size of the final kernel matrix, that needs to be inverted, is independent of the resolution because we use a group pooling or SumPool layer, respectively. Since the analytic solution in (41) only applies for MSE loss, we constructed target vectors $\mathcal{Y} = \{y_0, \dots, y_N\}$ from classes $c$ according to $e_c - \frac{1}{9}\mathbf{1}$ as is standard in the NTK literature (Lee et al., 2020).

The CNN and GCNN architectures that were used are shown in Table 1. Note that the infinite-width limit refers to the number of channels, which is why we only need to specify the kernel sizes. The same training and test data was used for both models with a test data size of 1000 images. Both architectures have been implemented in the neural-tangents package (Novak et al., 2020).

## F.3. Molecular Energy Regression

We used the same kernel method resulting from the infinite-width and infinite-time limit as explained in Section F.2. Both the grid on $S^2$ as well as on SO(3) are equiangular Driscoll & Healy grids (Driscoll & Healy, 1994) with resolution $2L \times (2L-1)^4$ on $S^2$ and $(2L-1) \times 2L \times (2L-1)$ on SO(3) (parametrized in Euler angles). $L$ is the corresponding bandlimit defining the cutoff in the Fourier domain, i.e. only Fourier coefficients $l < L$ are considered. The input signals are sampled for $L = 6$.

As the labels we have used the internal energies $U_0$ of the molecules at $0\,\mathrm{K}$ after substracting the atomic reference energies. The hyperparameter $\beta$ in (42) was chosen as described in (Esteves et al., 2023) according to

$$\beta = \frac{\cos(\pi/4)) - 1)^2}{\log(0.05)} \tag{188}$$

---

[4]In the original work by (Driscoll & Healy, 1994) the grid contained actually $2L \times 2L$ points, but we have adapted our grid to the convention used in the s2fft package.

*Table 2.* Architectures used for the molecular energy regression described in Section 6. 29 identical networks (encaptured by curly braces) process the inputs associated to each atom. Their outputs are then summed together. For group-convolutional and lifting layers, the output bandlimit $L$ is stated. The pooling layer is global and the single output neuron represents the predicted energy of the network.

| MLP | GCNN |
|---|---|
| *29 per-atom networks* | *29 per-atom networks* |
| $\left\{\begin{matrix}\texttt{Dense}\\\texttt{ReLU}\\\texttt{Dense}\end{matrix}\right\}$ ... $\left\{\begin{matrix}\texttt{Dense}\\\texttt{ReLU}\\\texttt{Dense}\end{matrix}\right\}$ | $\left\{\begin{matrix}\texttt{Lifting(3)}\\\texttt{Erf}\\\texttt{GConv(3)}\end{matrix}\right\}$ ... $\left\{\begin{matrix}\texttt{Lifting(3)}\\\texttt{Erf}\\\texttt{GConv(3)}\end{matrix}\right\}$ |
| *combined to molecule network* | *combined to molecule network* |
| `FanInSum` | `FanInSum` |
| `Dense(1)` | `Dense(1)` |

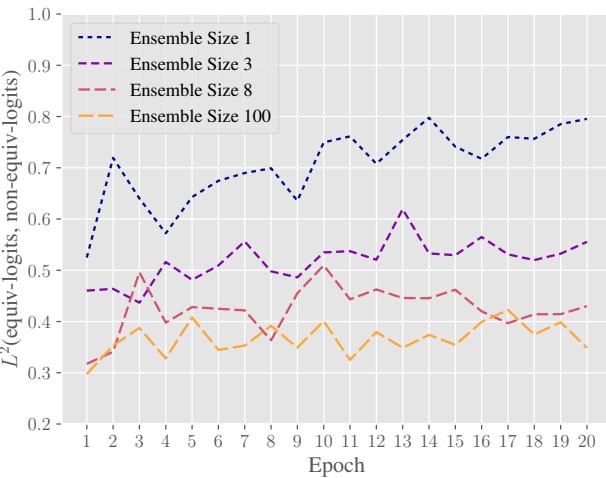

*Figure 6.* **Convergence of finite-width ensembles trained with data augmentation to ensembles of GCNNs on CIFAR10.** Shown is the $L^2$-distance between the logits of the equivariant ensemble and the non-equivariant ensemble trained with data augmentation for different ensemble sizes on out of distribution data. For larger ensembles, the distance decreases.

The precise architectures of the MLP based network and the $SO(3)$-invariant network are listed in Table 2. The MAE loss was evaluated on a test set of 100 molecules.

### F.4. Data Augmentation Versus Group Convolutions at Finite Width

Figure 6 shows that large ensembles trained with data augmentation on CIFAR10 converge to GCNNs even out of distribution. Similarly, Figure 7 shows the same behavior on the NCT-CRC-HE-100K data set of histological images (Kather et al., 2018), downscaled to $32 \times 32$ pixels. Samples of the out of distribution data, whose mean and variance were normalized to 0 and 1, respectively, are provided in Figure 8. The architectures used for the ensemble members are detailed in Table 3.

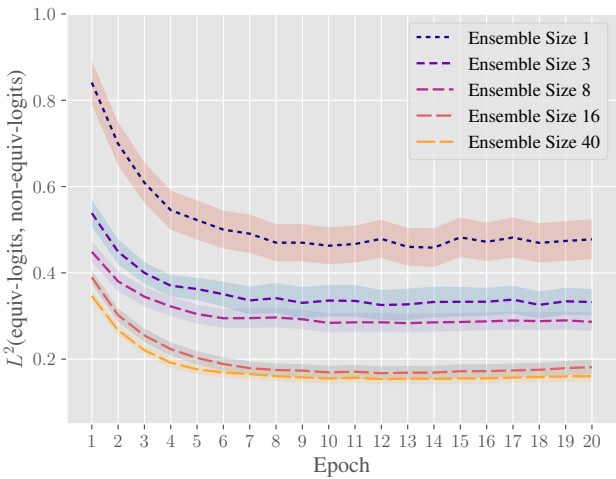

*Figure 7.* **Convergence of finite-width ensembles trained with data augmentation to ensembles of GCNNs on histological images from the NCT-CRC-HE-100K data set.** Shown is the $L^2$-distance between the logits of the equivariant ensemble and the non-equivariant ensemble trained with data augmentation for different ensemble sizes on out of distribution data. For larger ensembles, the distance decreases. The standard deviation is estimated from 20 independent runs for each curve.

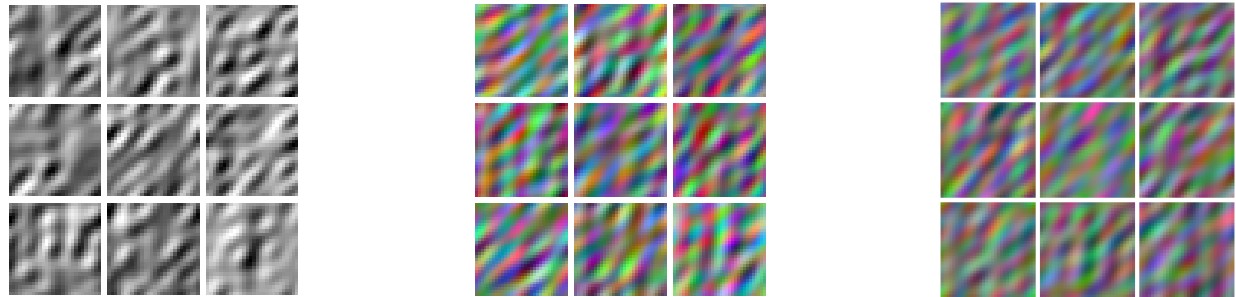

*Figure 8.* Examples for out of distribution data for MNIST (left) and CIFAR10 (middle) and NCT-CRC-HE-100K (right).

*Table 3.* Architectures used for the ensemble members in the experiments described in Section 6. For convolutional, group-convolutional and lifting layers, the arguments are input channels, output channels and kernel size (all kernels are squared). For max-pooling layers, the arguments are kernel size and stride. For the GCNNs, the max pooling is done only over spatial dimensions, not group dimensions. The kernel sizes were selected such that the GCNNs are exactly equivariant for the respective input sizes of $28 \times 28$ and $32 \times 32$.

| MNIST | | CIFAR10 | |
|---|---|---|---|
| CNN | GCNN | CNN | GCNN |
| Conv(1, 4, 3) | Lifting(1, 4, 3) | Conv(3, 4, 3) | Lifting(3, 4, 3) |
| ReLU | ReLU | ReLU | ReLU |
| MaxPool(2, 2) | SpatialMaxPool(2, 2) | MaxPool(2, 2) | SpatialMaxPool(2, 2) |
| Conv(4, 16, 4) | GConv(4, 16, 4) | Conv(4, 16, 4) | GConv(4, 16, 4) |
| ReLU | ReLU | ReLU | ReLU |
| MaxPool(2, 2) | SpatialMaxPool(2, 2) | MaxPool(2, 2) | SpatialMaxPool(2, 2) |
| Conv(16, 32, 3) | GConv(16, 32, 3) | Conv(16, 32, 3) | GConv(16, 32, 3) |
| ReLU | ReLU | ReLU | ReLU |
| Conv(32, 64, 3) | GConv(32, 64, 3) | Conv(32, 64, 4) | GConv(32, 64, 4) |
| ReLU | ReLU | ReLU | ReLU |
| Conv(64, 128, 1) | GConv(64, 128, 1) | Conv(64, 128, 1) | GConv(64, 128, 1) |
| ReLU | ReLU | ReLU | ReLU |
| Conv(128, 32, 1) | GConv(128, 32, 1) | Conv(128, 32, 1) | GConv(128, 32, 1) |
| ReLU | ReLU | ReLU | ReLU |
| Conv(32, 10, 1) | GConv(32, 10, 1) | Conv(32, 10, 1) | GConv(32, 10, 1) |
| | GPool | | GPool |

*Table 4.* Architectures used for the ensemble members in the experiments described in Section F.4. For convolutional, group-convolutional and lifting layers, the arguments are input channels, output channels and kernel size (all kernels are squared). For max-pooling layers, the arguments are kernel size and stride. For the GCNNs, the max pooling is done only over spatial dimensions, not group dimensions. The kernel sizes were selected such that the GCNNs are exactly equivariant for the input size of $32 \times 32$.

| NCT-CRC-HE-100K | |
|---|---|
| CNN | GCNN |
| Conv(3, 4, 3) | Lifting(3, 4, 3) |
| ReLU | ReLU |
| MaxPool(2, 2) | SpatialMaxPool(2, 2) |
| Conv(4, 16, 4) | GConv(4, 16, 4) |
| ReLU | ReLU |
| MaxPool(2, 2) | SpatialMaxPool(2, 2) |
| Conv(16, 32, 3) | GConv(16, 32, 3) |
| ReLU | ReLU |
| Conv(32, 64, 4) | GConv(32, 64, 4) |
| ReLU | ReLU |
| Conv(64, 128, 1) | GConv(64, 128, 1) |
| ReLU | ReLU |
| Conv(128, 32, 1) | GConv(128, 32, 1) |
| ReLU | ReLU |
| Conv(32, 9, 1) | GConv(32, 9, 1) |
| | GPool |

