# OpenReview forum: "Equivariant Neural Tangent Kernels"
_ICML.cc/2025/Conference — ICML 2025 poster_

### Official Review · Reviewer_kzsw · 2025-03-10

**Overall Recommendation:** 3

**Summary:**

This paper derives neural tangent kernels (infinite width) for group convolutional networks, and prove a equivalence between data augmentation and equivariance. They provide experimental validations for the finite-width framework. They also test group-equivariant kernels and compare them to non-equivariant ones.

## update after rebuttal

I thank the authors for their rebuttal. I will keep my score since In Th. 5.2 is about invariance, and not equivariance. I feel that a result on equivariance would be needed to recommend strong acceptance, since this paper is title Equivariant NTK.

**Claims And Evidence:**

The claim: “we show an interesting relationship between data augmentation and group convolutional networks. Specifically, we prove that they share the same expected prediction at all training times and even off-manifold” is a bit hard to understand.

Specifically: what do the authors mean by “expected” prediction: averaged over initializations? or over data distribution? Additionally, what is meant with “off-manifold”? In the main text, the authors refer to “ensembles” of neural networks, which I think is related. Could the authors clarify the framework and the claim there?

It seems that most results apply to equivariant neural networks with average pooling layers. How about max-pooling layers? Which of the authors’ results hold or do not hold here? If the results are restricted to average pooling, then the claims should be updated accordingly.

**Essential References Not Discussed:**

Taco Cohen’s dissertation.

**Experimental Designs Or Analyses:**

I did not see any issue.

**Methods And Evaluation Criteria:**

Yes, the datasets and benchmarks are well-suited.

**Other Comments Or Suggestions:**

Some typos, eg We proof -> We prove.

Theorem 4.2: if it’s the lifting layer, then we should have (1) and (2) instead of (l) and (l+1)?

Eq (8): It cannot be the same rho acting on f or on N(f), since f and N(f) do not have the same support: f’s support is not necessarily G whereas N(f) has support G.

**Other Strengths And Weaknesses:**

The introduction to NTK could be though to follow for someone not familiar with the topic. I would also recommend *not* starting with a reference to the appendix: the main text should be self-contained without readers having to go to the appendix. Examples of clarification point: why is it called the “frozen” NTK?

**Questions For Authors:**

NA.

**Relation To Broader Scientific Literature:**

Missing Taco Cohen’s dissertation.

Could the authors discuss an extension to non-regular group-equivariant NNs (eg, steerable?): would the results generalize to this case: why or why not?

How about the equivariant architectures from Finzi et al. A Practical Method for Constructing Equivariant Multilayer Perceptrons for Arbitrary Matrix Groups?

**Theoretical Claims:**

For the theorems, not much is said about the number of layers. Does it matter? Do the equivariant and non-equivariant architectures have the same amount?

In Th. 5.1. Is the assumption of “being related with group averaging” restrictive for the equivariant neural network?

In Th. 5.2, the N^GC is G-invariant : does the theorem apply for a G-equivariant neural network? Why restricting to invariance here?

---

> ### Author Rebuttal · Authors · 2025-04-01
>
> ## Expectation and “off-manifold”
>
> We are thankful for pointing out this confusion. The mentioned ensembles are a collection of independently initialized NNs and the average is understood over this family. For considerations about single networks, see heading “Ensembles” in the rebuttal for reviewer hpPR. The presented theorems do not make any particular assumptions over the data distribution.
>
> The term “off-manifold” refers to the *manifold hypothesis* of the data distribution. Our presented results hold for arbitrary inputs and hence, also hold off-manifold. We will clarify this in the updated manuscript.
>
> ## Average vs. max pooling
>
> Indeed our recursion relation in Theorem 4.3 only applies to average pooling. Since the expectation value and the maximum operation do not commute this is already true for MLPs and not a specific property of GCNNs.
>
> The average pooling which appears in Theorem 5.2 on the other hand is part of our result: Training an MLP with data augmentation corresponds at infinite width to a GCNN with a group average layer.
>
> ## Network depth in Theorems 5.2 and 5.3
>
> Indeed, both Theorem 5.2 and 5.3 hold for arbitrary number of layers. For example, the equivalence between the fully connected network defined in (33) and the corresponding GCNN defined in (34) holds for any number of layers $L$. We will add a clarifying remark.
>
> ## Assumption of NTK-relation in Theorem 5.1
>
> Theorem 5.1 yields a condition on the NTKs to relate augmented to non-augmented networks. The group average on the right hand side of (32) is a consequence of the data augmentation. Theorems 5.2 and 5.3 then provide particular architectures that satisfy this condition.
>
> ## Extension to equivariance
>
> To extend our results to equivariant networks, one would first need to extend Theorem 5.1 to equivariant augmentation. This is indeed possible by considering a network $\mathcal{N}$ which maps feature maps $f:X\rightarrow\mathbb{R}^n$ into feature maps $\mathcal{N}(f):G\rightarrow\mathbb{R}^{n'}$ by optimizing $\mathcal{N}(\rho_{\text{reg}}(g)f)$ against targets $\rho_{\text{reg}}(g)\hat{f}$ for all $g\in G$. The result is that the mean predictions agree if the NTKs of the two networks satisfy
>
> $$
> \Theta^{g,g'}(f,f')=\frac{1}{\mathrm{vol}(G)}\int_{G}\mathrm{d}h\Theta_{\mathrm{aug}}^{hg,g'}(\rho_{\text{reg}}(h)f,f').
> $$
>
> In order to find networks whose NTKs satisfy this relation, new layers need to be introduced since in the infinite-widht limit, the NTK of MLPs becomes proportional to the unit matrix in the output channels, trivializing the feature map. Here, a separation of the channels into a part which is taken to infinity and a part which is kept finite, e.g.
>
> $$
> f'_i(g)=\sum\_{j=1}^{n_c}\int\_X\mathrm{d}x\kappa\_{ij}(g,x)f_j(x), \qquad n_c \to \infty,
> $$
>
> which corresponds to a fully-connected layer at finite width, could be suitable. We haven’t repeated the calculations for this layer, but it would be an interesting extension of our results which we will mention in the manuscript.
>
> ## Non-regular representations
>
> Our analysis focuses on group convolutions. These can also be used for (scalar) point clouds by defining the input feature map to be $f(x)=\sum_i \delta(x-x_i)f_i$ for features $f_i$ at $x_i$. Then, the GCNN layers are equivariant with respect to transformations of the positions of the input features. However, this does not cover features transforming in vector- or tensor representations. For these, the setup would need to be extended. Although we have not performed this extension, we do not see conceptual roadblocks at the moment.
>
> ## Relation to Finzi et al.
>
> In contrast to this work, we specialize to the regular representation. This has the advantage that we can use the convolution theorem to compute the kernel recursions in Fourier space. Extending our framework to arbitrary representations would be a very interesting. A starting point would be to define suitable infinte-width limits of the relevant layers, see e.g. the section “Extension to equivariance” above.
>
> ## Introduction to the NTK
>
> We acknowledge that NTKs are a topic of substantial technical difficulty. We will revise the background section to make it more accessible stressing that the additional section in the appendix is optional background material. We will also explain the term “frozen” NTK that was adapted from other references (see e.g. Mario Geiger et al. [J. Stat. Mech. 2020]). It refers to the fact that the NTK becomes time-independent during training in the infinite width limit.
>
> ## Further remarks
>
> - We will add Taco Cohen’s PhD thesis to the literature review.
> - Thank you for pointing out the typos which we will fix in a revised version.
> - For full generality, we formulated a general layer-$\ell$ lifting layer, because one may include other non-group convolutional layers before as a preprocessing step.
> - We will clarify the notation in (8). Indeed, the regular representations on both sides act on functions with different domains.

---

### Official Review · Reviewer_hpPR · 2025-03-13

**Overall Recommendation:** 3

**Summary:**

This paper studies the training dynamics of equivariant neural networks via neural tangent kernels. The authors derive NTKs for group convolutions and nonlinearities, and also consider group convolutions for SO(3) in the Fourier domain (similar to spherical CNNs and G-steerable CNNs). The authors show that non-equivariant models with data augmentation converge to a specific GCNN architecture. Empirical results show that the NTK converges to the analytical expression for GCNNs. They also show that equivariant GCNNs outperform standard MLPs and data augmentation helps the standard MLPs converge to the same performance.

**Claims And Evidence:**

Yes the claims are specific and clearly supported mostly with proofs and also with empirical results.

**Essential References Not Discussed:**

No

**Experimental Designs Or Analyses:**

No

**Methods And Evaluation Criteria:**

Yes

**Other Comments Or Suggestions:**

None

**Other Strengths And Weaknesses:**

Strengths:
- Nice derivations of the NTKs for group convolutions
- Practical use case of the derived NTKs for comparison with data augmentation
- Experiments seem to practically support the paper contributions/claims
Weaknesses:
- See questions

**Questions For Authors:**

- Is the comparison with data augmentation and equivariant networks limited to ensembles only?
- As I am not familiar with NTKs, can the NTK for equivariant NNs describe the sample complexity improvements over standard NNs (perhaps the eigenvalues)? In the comparison with data augmentation, in my opinion, the greatest benefit of equivariant NNs over data augmentation is that they can achieve better performance with fewer data samples. E.g. a C_4 equivariant network only needs to see one sample while data augmentation with a standard network needs to see all 4 samples.  I can understand why data augmentation leads to a specific GCNN, but it would seem that GCNNs would be much more sample efficient.

**Relation To Broader Scientific Literature:**

The key contributions of this paper are deriving NTKs for group convolutions and using these results to show that a standard CNN on augmented data is equivalent to training an equivariant CNN on unaugmented data.

**Theoretical Claims:**

No, I did not check for correctness of any of the proofs.

---

> ### Author Rebuttal · Authors · 2025-04-01
>
> ## Ensembles
>
> Thank you for this important question. Although Theorem 5.1 is formulated in terms of ensembles, the statement in fact also holds for individual models at finite width if the infinite-width NTKs in (32) are replaced by the empirical NTKs
>
> $$
> \Theta(x,x')=\left(\frac{\partial\mathcal{N}(x)}{\partial\theta}\right)^\top \left(\frac{\partial\mathcal{N}(x')}{\partial\theta}\right)\,,
> $$
>
> as long as they are initialized in such a way that $\mathcal{N}(x)=\mathcal{N}^{\text{aug}}(x)=0$ $\forall x$ at initialization.
>
> Theorem 5.2 however only holds for infinite-width NTKs. Therefore, the combined statement of the equivalence of data augmentation and GCNNs holds also for individual models at infinite width, if $\mathcal{N}^{\text{FC}}(x)=\mathcal{N}^{\text{GC}}=0$ $\forall x$ at initialization.
>
> We will add a corresponding comment to the manuscript.
>
> ## Sample efficiency
>
> This is indeed an interesting line of further research for which our study lays the ground work. It is well known, see [arXiv:1912.13053], that the conditioning number of the NTK kernel is related to the convergence speed of training. One could therefore compare the conditioning number of the equivariant and standard NTKs to deduce insights into sample efficiency. This is however a task of significant technical difficulty (see Appendix B of the publication linked above which we’d need to generalize to the equivariant case) as it requires the analytical study of the NTK’s spectra as well as their corresponding phase structure which warrants follow-up work.

---

### Official Review · Reviewer_bkX8 · 2025-03-15

**Overall Recommendation:** 3

**Summary:**

In this work, the authors propose a way to understand the training dynamics of equivariant models by deriving neural tangent kernels for a broad class of equivariant architectures based on group convolutions.  For rototranslations in 2D and 3D, the authors show that equivariant NTKs outperform their non-equivariant counterparts as kernel predictors for CIFAR10, MNIST, medical image classification, and property prediction on the QM9 dataset.

The theoretical formulation consists of recursive relations for the NTK and
the NNGP for group convolutional layers, lifting layers, and group pooling layers. These allow efficient calculation of these kernels for arbitrary group convolutional architectures and thus provide the necessary tools to analytically study their training dynamics
in the large width limit.

Experimental claim: Networks trained with data augmentation converge to group convolutional networks at infinite width.

## post rebuttal

I wish to keep my score after reading the authors' rebuttal. I do not wish to update the score as the experiments still feel limitive ( although the authors do provide an additional experiment in the rebuttal)

**Claims And Evidence:**

Yes, they are clear. The one mentioned below is unclear and if the authors could provide clarification, that would be helpful.

For the experimental claim:  ln 419.(column 2) '.. data augmentation converge to group convolutional
networks at infinite width. We verify that this also holds
approximately at finite width',  but it does not seem fully justified, as it is a much simpler experiment.

**Essential References Not Discussed:**

Missed references:

1. On genuine invariance learning without weight-tying, Mokalev et al. [TAGML, ICML 23]

2. On the Ability of Deep Networks to Learn Symmetries from Data: A Neural Kernel Theory, Perin et al.

3. Data Augmentation vs. Equivariant Networks: A Theory of Generalization on Dynamics Forecasting, Wang et al. [ICMLworkshop 22]

4. Fast, Expressive Equivariant Networks through Weight-Sharing in Position-Orientation Space, Bekkers et al. [ICLR 24]

5. Spectrum Dependent Learning Curves in Kernel Regression and Wide Neural Networks, Borelon et al. [ICML 22]

6. Spectral bias and task-model alignment explain generalization in kernel regression and infinitely wide neural networks, Canatar et al. [Nat Comms 2021]

**Experimental Designs Or Analyses:**

Yes.

See weaknesses.

**Methods And Evaluation Criteria:**

The proposed experiments, datasets, and evaluation criteria make sense for the problem and method proposed.

**Other Comments Or Suggestions:**

Additional discussion/connection to missed papers in literature would be useful.

**Other Strengths And Weaknesses:**

## Strengths
- The paper is well motivated theoretically, and the experimental section supports the claims made in the paper.

- The paper is fairly well-written.



## Weaknesses

- The experiments for data augmentation vs finite width of neural networks are only shown for simpler toy datasets like CIFAR10 and MNIST

- Important citations and connections are missing.

**Questions For Authors:**

1.  What about comparing data augmentation and finite depth neural network work for more complex tasks like force prediction in QM9?

2. Elaborate on ln 419 (column 2) 'we proved that networks trained with data augmentation converge to group convolutional
networks at infinite width. We verify that this also holds approximately at finite width.' Explain what is considered approximate here.

3. ln 422. (column 1) Molecular energy regression. We know that energy is a scalar and an invariant quantity for a molecule with SO(3) transformations. Compared to a vector quantity like force prediction/regression, energy is a relatively easier task (as it is a scalar). Comment on the proposed claims in the paper in the light of different tasks based on its geometric complexity.

and see Weaknesses.

**Relation To Broader Scientific Literature:**

A few important citations are missed in the paper and thus, the connections to them are not made.
The key contributions to the paper are related to the broader literature; for example, the connection to spectral components and NTk have been made in literature. However, the GP formalism with equivariant NTK is relatively novel. Although claims in [1] are of a similar flavor as this paper, the theoretical formalism is different.

1. On the Ability of Deep Networks to Learn Symmetries from Data: A Neural Kernel Theory, Perin et al.

**Theoretical Claims:**

I have looked at the proof for Theorem 5.1 only. It has no issues.

---

> ### Author Rebuttal · Authors · 2025-04-01
>
> ## Approximate results at finite width
>
> Our theoretical claims hold for infinitely wide networks and in the ensemble mean, i.e. for infinitely large ensembles (for comments on extensions to single networks see rebuttal to reviewer hpPR, heading “Ensembles”). In this case, they predict exact agreement between the mean prediction of the invariant GCNN (due to group pooling) and the augmented network throughout training for arbitrary inputs. In our experiments, we test the agreement of the mean predictions for finite ensembles of finite-width networks and find that the predictions align more for larger ensembles. As expected, alignment is imperfect due to finite width, thus we call these results “approximate”. We will clarify this in the revised version.
>
> ## Extension to equivariant infinite width regression
>
> An extension of our framework to equivariant tasks like quantum mechanical force field prediction is indeed an interesting point. Using the GCNN layers we analyze, it is straightforward to perform regression on targets that are signals on $\mathrm{SO}(3)$, i.e. $f_\text{target}: \mathrm{SO}(3) \to \mathbb{R}^{n_\text{out}}$ in an equivariant way. Here, equivariance is understood with respect to the regular representation, see eq. (5). An equivariant model that is suitable for regression on vector quantities like forces necessitates an output layer that is equivariant with respect to the defining representation of $\mathrm{SO}(3)$. For this, the presented framework would need to be extended by additional layer types and their corresponding kernel relations. (One approach could be to take the argmax of the signal on $\mathrm{SO}(3)$ at the output layer and apply the resulting rotation to a fixed reference vector.) Furthermore, the infinite-width limit of the non-equivariant networks needs to be taken with care in this case. We have outlined a possible way to achieve this in the reply to reviewer *kzsw* under “Extension to equivariance”. Similarly as mentioned in the discussion by Cohen et al. [ICLR 2018], an extension to e.g. steerable CNNs would also be suitable approach. We consider this an interesting further research direction and will add a discussion to the manuscript.
>
> ## Complexity of the finite width experiments
>
> We recognize the interest in experiments with more demanding tasks. Since our theorems draw a connection between data augmented and invariant GCNN ensembles, we need to choose an invariant task to test the analytic results numerically. We have now repeated the experiment on a subset of the *NCT-CRC-HE-100K* data set (https://zenodo.org/records/1214456) consisting of histological images. We have trained on down-sampled images with a resolution of `32x32` pixels and produced OOD samples in this input space. The results are shown in  https://postimg.cc/hQzcRpkf, which were obtained in the same fashion as Figure 4 and 6. We use the same architecture as for the CIFAR10 experiment apart from twice as many channels in each hidden layer and we also adapted the learning rate. Each curve is computed 20 times, we plot the means for the metric over the runs and their standard deviations. The larger ensembles reach a validation accuracy of around 80%. The results further support our theoretical claims and we will add them to the manuscript.
>
> ## Citations
>
> > 1. On genuine invariance learning without weight-tying, Mokalev et al. [TAGML, ICML 23]
> >
> > 1. On the Ability of Deep Networks to Learn Symmetries from Data: A Neural Kernel Theory, Perin et al.
> > 2. Data Augmentation vs. Equivariant Networks: A Theory of Generalization on Dynamics Forecasting, Wang et al. [ICMLworkshop 22]
> > 3. Fast, Expressive Equivariant Networks through Weight-Sharing in Position-Orientation Space, Bekkers et al. [ICLR 24]
> > 4. Spectrum Dependent Learning Curves in Kernel Regression and Wide Neural Networks, Borelon et al. [ICML 22]
> > 5. Spectral bias and task-model alignment explain generalization in kernel regression and infinitely wide neural networks, Canatar et al. [Nat Comms 2021]
>
> We thank the reviewer for suggesting a number of relevant references and will add them to a revised version of the manuscript.
>
> ### Relation to the work by Perin et al. [arXiv:2412.11521]
>
> Thank you for pointing out Perin et al. We will include this very recent and interesting contribution in the revised manuscript. As you point out, although they study a similar problem, they focus on a very specific toy problem for which they analyze the spectrum of the NTK in detail. The only equivariant architecture they consider are CNNs. In contrast, our framework captures the learning dynamics of arbitrary group convolutional neural networks.

---

### Decision · Program_Chairs · 2025-05-01

**Decision:**

Accept (poster)

**Comment:**

The submission received unanimous positive, but lukewarm reviews. Reviewer bkX8 raised the most serious issue regarding the approximate convergence between data augmentation and GCNNs in the finite width case, which was only show in simple datasets and contradicts previous published results. This does not affect the main theoretical results for the infinite width case, including the interesting connection between data augmentation and GCNNs. The reviewer recommends acceptance despite this potential issue and I agree with their recommendation.

Reviewer bkX8 also pointed out missing citations and discussions, while kzsw raised small issues with the formality, which seemed adequately addressed in the rebuttal. I urge the authors to update the writing following the reviewers suggestions.